# Minimax-optimal and Locally-adaptive
# Online Nonparametric Regression

**Paul Liautaud**                           PAUL.LIAUTAUD@SORBONNE-UNIVERSITE.FR
*Sorbonne Université, CNRS, LPSM, F-75005 Paris, France*

**Pierre Gaillard**                          PIERRE.GAILLARD@INRIA.FR
*Université Grenoble Alpes, INRIA, CNRS, Grenoble INP, LJK, Grenoble, 38000, France*

**Olivier Wintenberger**          OLIVIER.WINTENBERGER@SORBONNE-UNIVERSITE.FR
*Sorbonne Université, CNRS, LPSM, F-75005 Paris, France*
*Institut Pauli CNRS, Vienna University, Oskar Morgenstern Platz 1, 1090 Wien, Austria*

**Editors:** Gautam Kamath and Po-Ling Loh

## Abstract

We study adversarial online nonparametric regression with general convex losses and propose a parameter-free learning algorithm that achieves minimax optimal rates. Our approach leverages chaining trees to compete against Hölder functions and establishes optimal regret bounds. While competing with nonparametric function classes can be challenging, they often exhibit local patterns - such as local Hölder continuity - that online algorithms can exploit. Without prior knowledge, our method dynamically tracks and adapts to different Hölder profiles by pruning a core chaining tree structure, aligning itself with local smoothness variations. This leads to the first computationally efficient algorithm with locally adaptive optimal rates for online regression in an adversarial setting. Finally, we discuss how these notions could be extended to a boosting framework, offering promising directions for future research.

**Keywords:** Online Learning, Local Adaptivity, Nonparametric Regression

## 1. Introduction

Observing a stream of data $x_1, x_2, \ldots$, an online regression algorithm sequentially predicts a function $\hat{f}_t \in \mathbb{R}^{\mathcal{X}}$ at each time step $t \geqslant 1$ based on the current input $x_t \in \mathcal{X} \subset \mathbb{R}^d$, where $d \geqslant 1$. The accuracy of these predictions is measured using a sequence of convex loss functions $(\ell_t)_{t \geqslant 1}$. Examples include the absolute loss $\ell_t(\hat{y}) = |\hat{y} - y_t|$ and the squared loss $(\hat{y} - y_t)^2$, for some observation $y_t \in \mathbb{R}$. The performance of an online regression algorithm is evaluated through its *regret* relative to a competitive class of functions $\mathcal{F} \subset \mathbb{R}^{\mathcal{X}}$, defined over a time horizon $T \geqslant 1$ as

$$\text{Reg}_T(f) := \sum_{t=1}^{T} \ell_t(\hat{f}_t(x_t)) - \sum_{t=1}^{T} \ell_t(f(x_t)), \qquad \forall f \in \mathcal{F}. \tag{1}$$

The function class $\mathcal{F}$ is typically chosen to capture smooth or structured relationships in the data, such as Lipschitz functions, which are commonly used to model nonparametric regression problems. Unlike traditional batch regression methods, which train models on the full dataset $\{(x_s, \ell_s)\}_{s=1}^{T}$, online regression algorithms - see Cesa-Bianchi and Lugosi (2006) for a reference textbook - make predictions sequentially, updating $\hat{f}_t$ at each step using only past observations $\{(x_s, \ell_s)\}_{s=1}^{t-1}$. This sequential and adaptive learning paradigm allows algorithms to capture complex and evolving patterns in the data without requiring strong assumptions, such as i.i.d. observations.

A fundamental goal in online regression is to design algorithms that are *minimax-optimal* in an adversarial setting, meaning they achieve the best possible regret guarantees over the worst-case data sequence - see Rakhlin and Sridharan (2014, 2015). Existing methods, such as those of Gaillard and Gerchinovitz (2015); Cesa-Bianchi et al. (2017), can attain minimax rates when the regularity of the functions in the competitive class is known beforehand, but they do not extend to cases where the function's smoothness varies across the domain, requiring a more flexible and adaptive strategy. Later, Kuzborskij and Cesa-Bianchi (2020) developed a locally adaptive algorithm, but it achieves a suboptimal regret rate. Thus, designing algorithms that adapt locally to unknown regularities and variations while maintaining optimal regret guarantees remains a key open challenge.

In this work, we propose a computationally efficient online learning algorithm that achieves *locally adaptive minimax regret* without requiring prior knowledge of the competitor's regularity. Our method builds on chaining trees and leverages an adaptive pruning mechanism that dynamically adjusts to local smoothness variations in the competitor function. Inspired by prior work on tree-based online learning (Kuzborskij and Cesa-Bianchi, 2020), we introduce a core tree structure that selects prunings in an optimal way, ensuring adaptivity to different Hölder profiles. This leads to the first online regression algorithm that is both minimax-optimal and locally adaptive, bridging the gap between computational efficiency, minimax rates, and local adaptivity. Additionally, our algorithm is general and applies to both convex and exp-concave loss functions, achieving optimal regret guarantees under mild assumptions. Finally, we validate our theoretical results with numerical experiments[1] demonstrating the practical benefits of our approach.

As a conclusion and perspective, we highlight how our approach shares similarities with boosting's iterative refinement process and discuss how this connection could inspire future work in online regression.

## 1.1. Related work

### 1.1.1. ONLINE NONPARAMETRIC REGRESSION

Vovk (2006) introduced online nonparametric regression with general function classes. Cesa-Bianchi and Lugosi (2006) developed an algorithm that exploits loss functions with good curvature properties, such as exp-concavity, to achieve fast regret rates in adversarial settings. Rakhlin and Sridharan (2014) further advanced the minimax theory, providing a non-polynomial algorithm that is optimal for regret in cumulative squared errors of prediction. This theory was later extended to general convex losses in Rakhlin and Sridharan (2015). A significant step toward computational efficiency was made by Gaillard and Gerchinovitz (2015); Cesa-Bianchi et al. (2017) designed a polynomial-time chaining algorithm that achieves minimax regret when the regularity of the competitor is known. They also observed that the same algorithm, with a different tuning, remains minimax-optimal for general convex losses.

In the batch statistical setting with i.i.d. data, the convergence rates of tree-based aggregation methods have been primarily studied in the context of random forests; see Biau and Scornet (2016) for a survey. Avoiding early stopping and overfitting, the purely random forests of Arlot and Genuer (2014) achieve minimax rates for i.i.d. nonparametric regression. Closer to our setting, Mourtada et al. (2017) studied the aggregation of Mondrian trees trained sequentially but in a batch (i.i.d.) statistical framework. While their method adapts to the regularity of the unknown regression function in well-specified settings, it does not extend to adversarial environments.

---

1. The code to reproduce all numerical experiments can be found here.

| References | Assumptions | Upper bound |
|:---:|:---|:---:|
| This paper | $(\ell_t)$ exp-concave, $L > 0$ unknown $(\ell_t)$ convex, $L > 0$ unknown | $\min\left\{\sqrt{LT}, L^{\frac{2}{3}}T^{\frac{1}{3}}\right\}$ $\sqrt{LT}$ |
| KCB20 | $(\ell_t)$ square loss, $L > 0$ unknown | $\sqrt{LT}$ |
| HM07 | $(\ell_t)$ absolute loss, $L > 0$ known $(\ell_t)$ square loss, $L > 0$ known | $L^{\frac{1}{3}}T^{\frac{2}{3}}$ $\sqrt{LT}$ |
| GG15 | $(\ell_t)$ square loss, $L = 1$ known | $T^{\frac{1}{3}}$ |
| CB3G17 | $(\ell_t)$ convex, $L = 1$ known | $\sqrt{T}$ |

Table 1: Comparison of regret guarantees for recent algorithms in online nonparametric regression with dimension $d = 1$ and smoothness rate $\alpha = 1$.

### 1.1.2. REGRET AGAINST $\alpha$-HÖLDER COMPETITORS AND LOCAL ADAPTIVITY

Considering $\mathcal{F}$ as the set of Lipschitz functions ($\alpha = 1$ in Equation (5)) for any constant $L > 0$, Hazan and Megiddo (2007) introduced the corresponding minimax regret. They proved that for $d = 1$, the minimax rate is $O(\sqrt{LT})$ for any convex loss, motivating the design of an algorithm that localizes at an optimal rate depending on $L$. The knowledge of $L$ is crucial for their procedures to prevent regret from growing linearly with $L$. Going one step further, Kuzborskij and Cesa-Bianchi (2020) demonstrated the adaptability of tree-based online algorithms by introducing an oracle pruning procedure in the regret analysis, given a core tree. Tracking the best pruning goes back to Helmbold and Schapire (1995) and Margineantu and Dietterich (1997). Kpotufe and Orabona (2013) also designed adaptive pruning algorithms based on trees to partition the instance space $\mathcal{X}$ optimally and sequentially. Competing with an oracle pruning in nonparametric regression enables adaptation to the local regularities $(L, \alpha)$ of $\alpha$-Hölder continuous functions - see (5). Indeed, the implicit multi-resolution nature of pruning allows the depth of the leaves to align with local Hölder constants: the larger the constant, the deeper the pruning, as finer partitions are needed to capture variations in the function.

However, existing methods either require prior knowledge of local Hölder constants in and the exponent rate $\alpha$, or they fail to attain minimax regret rates in polynomial time. The problem of designing a computationally efficient, minimax-optimal algorithm that adapts to local regularities remained open. One solution to this problem relies on an adaptive pruning approach on chaining trees, dynamically tracking and aligning with different Hölder profiles to adjust the depth of partitioning in an online manner. We propose an algorithm that efficiently adapts to local smoothness variations (both in $L$ and $\alpha$) without requiring prior knowledge of the underlying function regularities. Table 1 presents an overview of our main result alongside previous advancements in the field of online (and local) nonparametric regression.

## 1.2. Contributions and outline of the paper

We first present, in Section 2, a parameter-free online learning method that leverages a chaining tree structure and achieves minimax regret over $\alpha$-Hölder continuous functions with global exponent rate $\alpha \leqslant 1$. Next, in Section 3, we introduce a core tree adaptive algorithm that dynamically tracks

and adjusts to local smoothness variations through an adaptive pruning mechanism, enabling it to efficiently compete against functions with different local regularities. We prove that our approach achieves an optimal locally adaptive regret bound in an adversarial setting. In particular, we show that our algorithm adapts to the curvature of the loss functions and remains optimal for both general convex and exp-concave losses.

Finally, we include numerical experiments in the supplementary materials (Appendix F) to illustrate our results on a synthetic dataset. As interesting perspectives, we also draw connections between our approach and boosting techniques, suggesting a potential foundation for a boosting theory in adversarial online regression.

## 2. Minimax regret with chaining trees: a parameter-free online approach

**Setting and notations.** We consider that data $x_1, x_2, \cdots \in \mathcal{X}$ arrive in a stream. At each time step $t \geqslant 1$, the algorithm updates $\hat{f}_t$, receives $x_t \in \mathcal{X}$ and predicts $\hat{f}_t(x_t) \in \mathbb{R}$. Then, a loss function $\ell_t : \mathbb{R} \to \mathbb{R}$ is disclosed. The learner incurs loss $\ell_t(\hat{f}_t(x_t))$ and considers gradients to update strategies for time $t + 1$. We assume that $(\ell_t)$ are convex, $G$-Lipschitz and attain one minimum within $[-B, B]$, for some $B > 0$. The input space $\mathcal{X}$ is a bounded subspace of $\mathbb{R}^d$, $d \geqslant 1$. We write $|\mathcal{X}'| = \sup_{x,x' \in \mathcal{X}'} \|x - x'\|_\infty < \infty$ for any $\mathcal{X}' \subset \mathcal{X}$ and $[N] = \{1, \ldots, N\}$ for $N \geqslant 1$.

In this section, we present our first contribution: an online learning algorithm (Algorithm 1) that leverages a specialized decision tree structure, referred to as chaining trees, which we introduce in the next section. Specifically, we establish in Theorem 1 that our procedure achieves minimax-optimal regret in nonparametric regression over the class of Hölder-continuous functions.

### 2.1. Chaining tree

Tree-based methods are conceptually simple yet powerful - see Breiman et al. (2017). They consist in partitioning the feature space into small regions and then fitting a simple model in each one. Given $\mathcal{X} \subset \mathbb{R}^d$, a *regular decision tree* $(\mathcal{T}, \bar{\mathcal{X}}, \bar{\mathcal{W}})$ is composed of the following components:

- a finite rooted ordered regular tree $\mathcal{T}$ of degree $\deg(\mathcal{T})$, with nodes $\mathcal{N}(\mathcal{T})$ and leaves or terminal nodes $\mathcal{L}(\mathcal{T}) \subset \mathcal{N}(\mathcal{T})$. The root and depth of $\mathcal{T}$ are respectively denoted by $\mathrm{root}(\mathcal{T})$ and $\mathrm{d}(\mathcal{T})$. Each interior node $n \in \mathcal{N}(\mathcal{T}) \backslash \mathcal{L}(\mathcal{T})$ has $\deg(\mathcal{T})$ children. The parent of a node $n$ is referred to as $\mathrm{p}(n)$ and its depth as $\mathrm{d}(n)$;

- a family of sub-regions $\bar{\mathcal{X}} = \{\mathcal{X}_n, n \in \mathcal{N}(\mathcal{T})\}$ consisting of subsets of $\mathcal{X}$ such that for any interior node $n$, $\{\mathcal{X}_m : \mathrm{p}(m) = n\}$ forms a partition of $\mathcal{X}_n$;

- a family of prediction functions $\bar{\mathcal{W}} = \{h_n : \mathcal{X} \to \mathbb{R}, n \in \mathcal{N}(\mathcal{T})\}$ associated to each node such that $h_n(x) = 0$ for all $x \notin \mathcal{X}_n$.

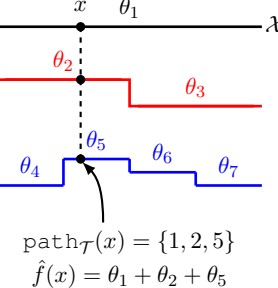

$$\mathrm{path}_\mathcal{T}(x) = \{1, 2, 5\}$$
$$\hat{f}(x) = \theta_1 + \theta_2 + \theta_5$$

Figure 1: Example of a CT over $\mathcal{X} \subset \mathbb{R}$.

The standard method of Breiman et al. (2017) for predicting with a decision tree is to use the partition induced by the leaves $\sum_{n \in \mathcal{L}(\mathcal{T})} h_n(x)$, $x \in \mathcal{X}$. On the contrary, the chaining tree that we define below, preforms multi-scale predictions by combining the predictions from all nodes.

**Definition 1 (Chaining-Tree)**  *A Chaining-Tree (CT) prediction function $\hat{f}$ is defined as*

$$\hat{f}(x) = \sum_{n \in \mathcal{N}(\mathcal{T})} h_n(x), \qquad x \in \mathcal{X} \subset \mathbb{R}^d,$$

*where the regular prediction tree $(\mathcal{T}, \bar{\mathcal{X}}, \bar{\mathcal{W}})$ satisfies:*
- *the prediction functions $h_n$ are constant $h_n(x) = \theta_n \mathbb{1}_{x \in \mathcal{X}_n}$, $\theta_n \in \mathbb{R}$. We denote them by $\theta_n$ by abuse of notation;*
- *the degree $\deg(\mathcal{T}) = 2^d$ and for any interior node $n$, $\{\mathcal{X}_m : \mathrm{p}(m) = n\}$ forms a regular partition of $\mathcal{X}_n$ in infinite norm. In particular, this implies $|\mathcal{X}_m| = |\mathcal{X}_{\mathrm{p}(m)}|/2$.*

We provide a schematic illustration in Figure 1. Chaining trees are closely related to the chaining technique introduced by Dudley (1967), which is at the core of algorithms addressing function approximation tasks. This method involves a sequential refinement process, that is - roughly speaking - growing a sequence of refining approximations over a function space. It was first introduced to design concrete online learning algorithm with optimal rates by Gaillard and Gerchinovitz (2015).

### 2.2. First algorithm: the online training of a chaining-tree

We introduce in this section an explicit Algorithm 1 to sequentially train our CT $\mathcal{T}$ over time.

---
**Algorithm 1:** Training CT $\mathcal{T}$ at time $t \geqslant 1$

---
**Input**  : $(\theta_{n,t})_{n \in \mathcal{N}(\mathcal{T})}$ (node predictors of $\mathcal{T}$), $(g_{n,t})_{n \in \mathcal{N}(\mathcal{T})}$ (gradients - later specified).
**for** $n \in \mathcal{N}(\mathcal{T})$ **do**

> Predict $\hat{f}_t(x_t) = \sum_{n \in \mathcal{N}(\mathcal{T})} \theta_{n,t} \mathbb{1}_{x_t \in \mathcal{X}_n}$;
> Find $\theta_{n,t+1} \in \mathbb{R}$ to approximately minimize
>
> $$\theta_n \mapsto \ell_t(\hat{f}_{-n,t}(x_t) + \theta_n \mathbb{1}_{x_t \in \mathcal{X}_n}) \quad \text{with} \quad \hat{f}_{-n,t}(x_t) = \hat{f}_t(x_t) - \theta_{n,t} \mathbb{1}_{x_t \in \mathcal{X}_n} \qquad (2)$$
>
> using gradient $g_{n,t} = \left[ \frac{\partial \ell_t\left(\hat{f}_{-n,t}(x_t) + \theta_n \mathbb{1}_{x_t \in \mathcal{X}_n}\right)}{\partial \theta_n} \right]_{\theta_n = \theta_{n,t}}$.

**end**
**Output** : $(\theta_{n,t+1})_{n \in \mathcal{N}(\mathcal{T})}$

---

To keep things concise, the gradient minimization step in (2) is expressed as:

$$\theta_{n,t+1} \leftarrow \texttt{grad-step}(\theta_{n,t}, g_{n,t}). \qquad (3)$$

where the function $\texttt{grad-step}(\theta, g)$ stands for *any* rule that updates $\theta \in \mathbb{R}$ from time $t$ to $t+1$ using some gradient $g \in \mathbb{R}$.

**Computation of the gradients.**   At each time $t \geqslant 1$ and for each node $n \in \mathcal{N}(\mathcal{T})$, the subgradient $g_{n,t}$ of the last loss $\ell_t(\hat{f}_t(x_t))$ with respect to $\theta_{n,t}$ can be computed explicitly using the chain rule:

$$g_{n,t} = \left[ \frac{\partial \ell_t(\hat{f}_{-n,t}(x_t) + \theta \mathbb{1}_{x_t \in \mathcal{X}_n})}{\partial \theta} \right]_{\theta = \theta_{n,t}} = \ell'_t(\hat{f}_t(x_t)) \mathbb{1}_{x_t \in \mathcal{X}_n}, \qquad n \in \mathcal{N}(\mathcal{T}), \qquad (4)$$

which simplifies the computation of subgradients, as the dependence on $n$ only involves the indicator function. More precisely, the subroutine $\texttt{grad-step}$, detailed below, does not perform any update

(i.e., $\theta_{n,t+1} = \theta_{n,t}$) when the gradient is zero (i.e., $x_t \notin \mathcal{X}_n$). All nonzero updates use the same subgradient $g_t = \ell'_t(\hat{f}_t(x_t))$, which is based on the derivative of the loss of the strong learner's prediction.

**Online gradient optimization subroutine.** We now detail the subroutine `grad-step`, which, in our analysis, can be any online optimization algorithm satisfying the following regret upper-bound.

**Assumption 1** *Let $g_{n,1}, \ldots, g_{n,T} \in [-G, G]$ for $T \geqslant 1$, $G > 0$, and $n \in \mathcal{N}(\mathcal{T})$. We assume that the parameters $\theta_{n,t}$ starting at $\theta_{n,1} \in \mathbb{R}$ and following the update (3) satisfy the linear regret bound:*

$$\sum_{t=1}^{T} g_{n,t}(\theta_{n,t} - \theta_n) \leqslant |\theta_n - \theta_{n,1}|\left(C_1\sqrt{\textstyle\sum_{t=1}^{T} |g_{n,t}|^2} + C_2 G\right),$$

*for some $C_1, C_2 > 0$ and every $\theta_n \in \mathbb{R}$.*

Such an assumption is satisfied by so-called *parameter-free* online convex optimization algorithms, such as those described in Cutkosky and Orabona (2018); Mhammedi and Koolen (2020); Orabona and Pál (2016). Specifically, by considering only the time steps where the gradients are nonzero, $T_n = \{1 \leqslant t \leqslant T : g_{n,t} \neq 0\}$, their procedure entails $O(G|\theta_n|\sqrt{|T_n|})$. Note that the constants $C_1, C_2$ often hide logarithmic factors in $T, G$ or $|\theta_n|$. These algorithms require no parameter tuning (though some need prior knowledge of $G$) and provide a regret upper bound that automatically scales with the parameter norm $|\theta_n|$. This property is crucial in analyzing our CT, where each node is tasked with correcting the errors of its ancestors in a more refined subregion of the input space. This multi-resolution aspect of the predictions leads us to consider $\theta_n$ that approach zero as $d(n)$ increases.

**First result.** In the theorem below, we show that when resorting to such a subroutine into Algorithm 1, our results are minimax-optimal with respect to $\mathscr{C}^\alpha(\mathcal{X}, L)$ the class of $\alpha$-Hölder continuous functions over $\mathcal{X}$ defined with $L > 0$ and $\alpha \in (0, 1]$ by

$$\mathscr{C}^\alpha(L, \mathcal{X}) := \left\{f : \mathcal{X} \to \mathbb{R} : |f(x) - f(x')| \leqslant L\|x - x'\|_\infty^\alpha, \ x, x' \in \mathcal{X} \text{ and } \sup_{x \in \mathcal{X}} |f(x)| \leqslant B\right\},$$
$$(5)$$

with $B > 0$ such that $\ell_t$ has minimum lying in $[-B, B]$. We will refer to $L$ as the *Hölder constant* and $\alpha$ to as the *smoothness rate* or *exponent*.

**Theorem 1** *Let $T \geqslant 1$, $(\mathcal{T}, \bar{\mathcal{X}}, \bar{\mathcal{W}}_1)$ be a CT with $\mathcal{X}_{\text{root}(\mathcal{T})} = \mathcal{X}$, $\theta_{n,1} = 0$ for all $n \in \mathcal{N}(\mathcal{T})$ and $d(\mathcal{T}) = \frac{1}{d}\log_2 T$. Then, Algorithm 1 applied with a `grad-step` procedure satisfying Assumption 1 achieves the regret upper bound*

$$\sup_{f \in \mathscr{C}^\alpha(\mathcal{X}, L)} \text{Reg}_T(f) \leqslant GB(C_1\sqrt{T} + C_2) + GL|\mathcal{X}|^\alpha \begin{cases} \left(\Phi(\frac{d}{2} - \alpha)C_1 + 4C_2 + 1\right)\sqrt{T} & \text{if } d < 2\alpha, \\ \left(\frac{C_1}{d}\log_2 T + 4C_2 + 1\right)\sqrt{T} & \text{if } d = 2\alpha, \\ \left(\Phi(\frac{d}{2} - \alpha)C_1 + 4C_2 + 1\right)T^{1-\frac{\alpha}{d}} & \text{if } d > 2\alpha, \end{cases}$$

*for any $L > 0$ and $\alpha \in (0, 1]$, where $\Phi(u) = |2^u - 1|^{-1}$.*

The proof of Theorem 1 is postponed to Appendix A.

**Minimax optimality and adaptivity to $L$ and $\alpha$.** Note that the above rates are minimax optimal for online nonparametric regression with convex losses over $\mathscr{C}^{\alpha}(\mathcal{X}, L)$, as shown by Rakhlin and Sridharan (2015) that provides a non-constructive minimax analysis for this problem (see also Rakhlin and Sridharan (2014)). For the case of low-dimensional settings, where $d \leqslant 2\alpha$, our bound is in $O((B + L)\sqrt{T})$. However, it has been demonstrated in Rakhlin and Sridharan (2015) that faster rates $O(T^{\frac{1}{3}})$ can be attained when dealing with exp-concave losses. In the next section, we will address this by making our algorithm adaptive to the curvature of the loss functions. A similar chaining technique was applied by Gaillard and Gerchinovitz (2015) to design an algorithm with minimax rates for the square loss or Cesa-Bianchi et al. (2017) in the partial information setting. However, unlike these works, our Algorithm 1 does not require prior knowledge of neither $L$ nor $\alpha$ and automatically adapts to them. This is achieved through the use of *parameter-free* subroutines that satisfy Assumption 1 and automatically adapt to the norm of $\theta_n$.

**Comparison to standard adaptive OCO methods in $\mathbb{R}^{|\mathcal{N}(\mathcal{T})|}$.** A key point of Algorithm 1 is its *node-specific* descent, which differs from standard adaptive OCO optimizing a global parameter. For each node $n \in \mathcal{N}(\mathcal{T})$ we obtain a regret upper-bound of $O(|\theta_n|\sqrt{\sum_t |g_{n,t}|^2})$ yielding an overall regret in $O(\sum_n |\theta_n|\sqrt{\sum_t |g_{n,t}|^2})$, with $g_{n,t}$ defined as in (4). Notably, thanks to the structure of the chaining-tree, $g_{n,t} = 0$ when the data $x_t$ does not fall in the corresponding sub-region of node $n$ and this leads to an overall regret scaling as $O(G \sum_n |\theta_n|\sqrt{|T_n|})$ where $T_n$ is the set of time steps for which $g_{n,t} \neq 0$.

One may wonder whether Algorithm 1 could be reduced to an adaptive Online Mirror Descent (OMD) on a *global* parameter $\boldsymbol{\theta} = (\theta_n)_{n \in \mathcal{N}(\mathcal{T})} \in \mathbb{R}^{|\mathcal{N}(\mathcal{T})|}$. This would result in an estimation regret bound, for any $p, q \geqslant 1$ such that $\frac{1}{p} + \frac{1}{q} = 1$,

$$O\Big(\|\boldsymbol{\theta}\|_p \sqrt{\textstyle\sum_{t=1}^{T} \|\mathbf{g}_t\|_q^2}\Big) \quad \text{where} \quad \mathbf{g}_t = \nabla_{\boldsymbol{\theta}} \ell_t\big(\textstyle\sum_{n \in \mathcal{N}(\mathcal{T})} \theta_{n,t} \mathbb{1}_{x_t \in \mathcal{X}_n}\big) = (g_{n,t})_{n \in \mathcal{N}(\mathcal{T})},$$

with $g_{n,t}$ as in (4). Moreover, we have for $q \geqslant 2$

$$\sum_n |\theta_n| \sqrt{\textstyle\sum_t |g_{n,t}|^2} \leqslant \|\boldsymbol{\theta}\|_p \Big(\textstyle\sum_n \big(\sum_t |g_{n,t}|^2\big)^{\frac{q}{2}}\Big)^{\frac{1}{q}} \qquad\qquad \leftarrow \text{by Hölder's inequality}$$

$$\leqslant \|\boldsymbol{\theta}\|_p \Big(\textstyle\sum_t \big(\sum_n |g_{n,t}|^q\big)^{\frac{2}{q}}\Big)^{\frac{1}{2}} \quad \leftarrow \text{by Minkowski's inequality with } \frac{q}{2} \geqslant 1$$

$$= \|\boldsymbol{\theta}\|_p \sqrt{\textstyle\sum_t \|\mathbf{g}_t\|_q^2}. \tag{6}$$

Remarkably, (6) shows that our Algorithm 1 consistently achieves a lower regret compared to any global adaptive OMD subroutine for $q \geqslant 2$ - including adaptive version of OGD ($p = q = 2$) and of EG ($p = 1, q = \infty$).

Finally, in our analysis in Appendix A, Proof of Theorem 1, such an adaptive OCO method would result in an overall estimation regret of $O(\sqrt{T} \sum_n |\theta_n|)$. By grouping by the level of the CT $\mathcal{T}$ (see Equation (21)), with $|\theta_n| \propto 2^{-\alpha m}, m \in [\mathrm{d}(\mathcal{T})]$, we get a regret of $O(2^{-\alpha m}|\{n : \mathrm{d}(n) = m\}|\sqrt{T})$ for each level instead of $O(2^{-\alpha m}\sqrt{|\{n : \mathrm{d}(n) = m\}|T})$, which is insufficient to recover the same minimax rates.

**Complexity.** Although the formal definition of our algorithm requires constructing a decision tree with $|\mathcal{N}(\mathcal{T})| = 2^{\mathrm{d}(\mathcal{T})d} = T$ nodes, it remains tractable, similar to the approach in Gaillard and Gerchinovitz (2015). At each round, the input $x_t$ falls into one node per level of the tree constituting

$\mathrm{path}_{\mathcal{T}}(x_t) = \{n \in \mathcal{N}(\mathcal{T}) : x_t \in \mathcal{X}_n\}$, since $\{\mathcal{X}_n, \mathrm{d}(n) = m\}$ forms a partition of $\mathcal{X}$ for any depth $1 \leqslant m \leqslant \mathrm{d}(\mathcal{T})$ - see Figure 1 for schematic comprehension. Consequently, most subgradients in (4) are zero, and grad-step only needs to be called $\mathrm{d}(\mathcal{T}) = \frac{1}{d}\log_2 T$ times per round, each using the same gradient $g_t$. Thus, the loop in Algorithm 1 can be rewritten to explore only the nodes along $\mathrm{path}_{\mathcal{T}}(x_t)$, significantly reducing computational complexity. The overall space complexity is at most $O(|\mathcal{N}(\mathcal{T})|) = O(T)$. It can be improved noticing that nodes in the tree do not need to be created until at least one input falls into that node.

**Unknown input space.** In practice our procedure can be easily extended to the case where $\mathcal{X}$ is unknown beforehand and is sequentially revealed through new inputs $x_t \in \mathbb{R}^d$ (similarly to Kuzborskij and Cesa-Bianchi (2020)). This can be done either through a doubling trick (starting with $\mathcal{X} = [-1, 1]^d$ and restarting the algorithm with an increased diameter by at least a factor of 2 each time an input falls outside of the current tree) or by creating a new CT around $x_t$ that runs in parallel, whenever a new point $x_t$ falls outside the existing trees.

## 3. Optimal and locally adaptive regret in online nonparametric regression

In the previous section, we demonstrated that our Algorithm 1 achieves minimax regret $O(LT^{(d-\alpha)/d})$ compared to Hölder functions $\mathscr{C}^\alpha(\mathcal{X}, L)$. This bound scales linearly with the constant $L$ and raises the question of whether our approximation method could be adapted to fit subregions with lower variation. Our second contribution is an algorithm that adapts on the local Hölder profile of the competitor. For any $f \in \mathscr{C}^\alpha(\mathcal{X}, L), \alpha \in (0, 1], L > 0$, and some subset $\mathcal{X}_n \subset \mathcal{X}$, the local Hölder constant $L_n(f)$ satisfies

$$L_n(f) \leqslant L \quad \text{and} \quad |f(x) - f(x')| \leqslant L_n(f)\|x - x'\|_\infty^\alpha, \tag{7}$$

for every $x, x' \in \mathcal{X}_n$. Recall that we assume that for any $f \in \mathscr{C}^\alpha(\mathcal{X}, L), \sup_{x \in \mathcal{X}}|f(x)| \leqslant B$. We define $[\cdot]_B := \min(B, \max(-B, \cdot))$ the clipping operator in $[-B, B]$ and a uniform discretization grid $\Gamma$ with precision $\varepsilon = T^{-\frac{1}{2}}$ as the set of $K = \lceil 2B/\varepsilon \rceil$ constants

$$\Gamma := \{\gamma_k = -B + (k-1)\varepsilon, k = 1, \ldots, K\} \subset [-B, B].$$

**Locally adaptive algorithm.** We base our predictions on a combination of several regular decision tree predictions (see Section 2.1). The latter are sitting in nodes of a *core tree* $(\mathcal{T}_0, \bar{\mathcal{X}}, \bar{\mathcal{W}})$, with $\bar{\mathcal{W}} = \{(\hat{f}_{n,k})_{k=1}^K, n \in \mathcal{N}(\mathcal{T}_0)\}$. In our main Algorithm 2, referred to as *Locally Adaptive Online Regression*, the core tree $\mathcal{T}_0$ provides an average prediction at each time step $t \geqslant 1$ as follows:

$$\hat{f}_t(x_t) = \sum_{n \in \mathcal{N}(\mathcal{T}_0)} \sum_{k=1}^K w_{n,k,t}\hat{f}_{n,k,t}(x_t),$$

where, for each pair $(n, k) \in \mathcal{N}(\mathcal{T}_0) \times [K]$
- $\hat{f}_{n,k,\cdot}$ is a clipped predictor associated with a CT $\mathcal{T}_{n,k}$ (see Definition 1), rooted at $\mathcal{X}_{\mathrm{root}(\mathcal{T}_{n,k})} = \mathcal{X}_n \in \bar{\mathcal{X}}$ and starting at $\theta_{\mathrm{root}(\mathcal{T}_{n,k}),1} = \gamma_k \in \Gamma, \theta_{n',1} = 0$ for $n' \in \mathcal{N}(\mathcal{T}_{n,k}) \setminus \{\mathrm{root}(\mathcal{T}_{n,k})\}$;
- the weight $w_{n,k,t}$ adjust the contribution of the predictor $\hat{f}_{n,k,t}$ such that the sum of all weights over the tree satisfies $\sum_{n \in \mathcal{N}(\mathcal{T}_0)} \sum_{k \in [K]} w_{n,k,t} = 1$ at any time $t \geqslant 1$.

First, Algorithm 2 sequentially trains the weights $(w_{n,k})_{(n,k) \in \mathcal{N}(\mathcal{T}_0) \times [K]}$ using two key subroutines: weight and sleeping, both inspired by classical expert aggregation methods. Specifically, the weight$(\tilde{\mathbf{w}}, \tilde{\mathbf{g}})$ subroutine refers to any general algorithm updating weights $\tilde{\mathbf{w}}$ with a given gradient $\tilde{\mathbf{g}}$ and satisfies the following Assumption 2.

**Assumption 2** *Let $\tilde{\mathbf{g}}_1, \ldots, \tilde{\mathbf{g}}_T \in [-G, G]^{K \times |\mathcal{N}(\mathcal{T}_0)|}$, for $T \geqslant 1$ and $G > 0$. We assume that the weight vectors $\tilde{\mathbf{w}}_t$, initialized with a uniform distribution $\tilde{\mathbf{w}}_1$ and updated via* weight *in Algorithm 2, satisfy the following linear regret bound:*

$$\sum_{t=1}^T \tilde{\mathbf{g}}_t^\top \tilde{\mathbf{w}}_t - \tilde{g}_{n,k,t} \leqslant C_3 \sqrt{\log(K|\mathcal{N}(\mathcal{T}_0)|) \sum_{t=1}^T \left( \tilde{\mathbf{g}}_t^\top \tilde{\mathbf{w}}_t - \tilde{g}_{n,k,t} \right)^2} + C_4 G,$$

*for some constants $C_3, C_4 > 0$ and for every $n \in \mathcal{N}(\mathcal{T}_0)$, $k \in [K]$.*

Well-established aggregation algorithms, such as those from Gaillard et al. (2014), Koolen and Van Erven (2015), and Wintenberger (2017), exhibit such second-order linear regret bounds.

---

**Algorithm 2:** Locally Adaptive Online Regression

---

**Input** : A core regular tree $(\mathcal{T}_0, \bar{\mathcal{X}}, \bar{\mathcal{W}})$ with root $\mathcal{X}$, bounds $G, B > 0$.
Initial prediction functions $\hat{f}_{n,k,1} = \tilde{f}_{n,k,1} = \theta_{\text{root}(\mathcal{T}_{n,k}),1} \mathbb{1}_{x \in \mathcal{X}_n}$ associated to CT $\mathcal{T}_{n,k}, k \in [K], n \in \mathcal{N}(\mathcal{T}_0)$.
Initial uniform weights $\tilde{\mathbf{w}}_1 = (\tilde{w}_{n,k,1})_{n \in \mathcal{N}(\mathcal{T}_0), k \in [K]}$.

**for** $t = 1$ **to** $T$ **do**
    Receive $x_t$;
    $\mathcal{N}_t \leftarrow \text{path}_{\mathcal{T}_0}(x_t)$;
    $\mathbf{w}_t \leftarrow \text{sleeping}(\tilde{\mathbf{w}}_t, \mathcal{N}_t)$ ;
    Predict $\hat{f}_t(x_t) = \sum_{n \in \mathcal{N}_t} \sum_{k=1}^K w_{n,k,t} \hat{f}_{n,k,t}(x_t)$ ;
    # Update weights of $\mathcal{T}_0$
    Reveal gradient $\tilde{\mathbf{g}}_t = \nabla_{\tilde{\mathbf{w}}_t} \ell_t(\sum_{n \in \mathcal{N}_t} \sum_{k=1}^K \tilde{w}_{n,k,t} \hat{f}_{n,k,t}(x_t) + \sum_{n \notin \mathcal{N}_t} \sum_{k=1}^K \tilde{w}_{n,k,t} \hat{f}_t(x_t))$ ;
    Udpate $\tilde{\mathbf{w}}_{t+1} \leftarrow \text{weight}(\tilde{\mathbf{w}}_t, \tilde{\mathbf{g}}_t)$ ;
    **for** $n \in \mathcal{N}_t, k \in [K]$ **do**
        # Update CT $\mathcal{T}_{n,k}$
        Reveal gradient $g_{n,k,t} = \ell_t'(\tilde{f}_{n,k,t}(x_t))$ ;
        Update $\tilde{f}_{n,k,t}$ associated to CT $\mathcal{T}_{n,k}$ using Algorithm 1 with $g_{n,k,t}$ ;
        Clip local predictor as $\hat{f}_{n,k,t+1} = [\tilde{f}_{n,k,t+1}]_B$ ;
    **end**
**end**
**Output** : $\hat{f}_{T+1} = \sum_{n,k} w_{n,k,T+1} \hat{f}_{n,k,T+1}$

---

Since $\mathcal{T}_0$ partitions the input space $\mathcal{X}$, only a subset $\mathcal{N}_t$ of the nodes in $\mathcal{N}(\mathcal{T}_0)$ contributes to predictions at each round $t \geqslant 1$. The set of active nodes is determined by $\mathcal{N}_t \leftarrow \text{path}_{\mathcal{T}_0}(x_t)$, which maps the data point $x_t$ to the active nodes $\{n \in \mathcal{N}(\mathcal{T}_0) : x_t \in \mathcal{X}_n\}$. This structure mirrors the *sleeping experts* framework introduced by Freund et al. (1997); Gaillard et al. (2014), and we incorporate it as a sleeping subroutine in Algorithm 2. The weights $\mathbf{w}_t$ are computed using the sleeping$(\tilde{\mathbf{w}}_t, \mathcal{N}_t)$ subroutine, defined as follows for all $k \in [K]$ and $n \in \mathcal{N}(\mathcal{T}_0)$:

$$w_{n,k,t} = 0 \quad \text{if } n \notin \mathcal{N}_t, \qquad w_{n,k,t} = \frac{\tilde{w}_{n,k,t}}{\sum_{n' \in \mathcal{N}_t} \sum_{k'=1}^K \tilde{w}_{n',k',t}} \quad \text{otherwise.} \tag{8}$$

This ensures that only the active nodes are contributing to the average prediction.

Second, our Algorithm 2 also employs Algorithm 1 to independently train the CTs
$$\{\mathcal{T}_{n,k}, (n,k) \in \mathcal{N}(\mathcal{T}_0) \times [K]\}$$
that reside within $\mathcal{T}_0$. For each $(n,k) \in \mathcal{N}(\mathcal{T}_0) \times [K]$, $\mathcal{T}_{n,k}$ is initialized with $\theta_{\text{root}(\mathcal{T}_{n,k}),1} = \gamma_k$ and $\theta_{n',1} = 0$ for all $n' \in \mathcal{N}(\mathcal{T}_{n,k}) \setminus \{\text{root}(\mathcal{T}_{n,k})\}$, and is then updated at each time $t \geqslant 1$ via Algorithm 1 with a given gradient $g_{n,k,t}$. Then, the local predictors associated to $(\mathcal{T}_{n,k})$ are clipped in $[-B, B]$.

**Pruning as local adaptivity.** Pruning techniques are frequently employed in traditional statistical learning involving subtrees to reduce overfitting or simplify models. In this context, each pruned tree represents a localized profile corresponding to a partition of $\mathcal{X}$. Our Algorithm 2 strives to learn the oracle pruning strategy to compete effectively against any $\alpha$-Hölder continuous function.

**Definition 2 (Pruning)** *Let $(\mathcal{T}_0, \bar{\mathcal{X}}, \bar{\mathcal{W}})$ be some regular tree with $\bar{\mathcal{W}} = \{(\hat{f}_{n,k})_{k \in [K]}, n \in \mathcal{N}(\mathcal{T}_0)\}$. A pruning or pruned regular decision tree $(\mathcal{T}, \tilde{\mathcal{X}}, \tilde{\mathcal{W}})$ consists in a subtree, i.e. $\mathcal{N}(\mathcal{T}) \subset \mathcal{N}(\mathcal{T}_0)$, with root $\mathcal{X}_{\text{root}(\mathcal{T})} = \mathcal{X}_{\text{root}(\mathcal{T}_0)}$ and prediction functions $\tilde{\mathcal{W}} = \{\hat{f}_{n,k_n}, n \in \mathcal{N}(\mathcal{T}), k_n \in [K]\} \subset \bar{\mathcal{W}}$. It predicts, at each time $t \geqslant 1$,*
$$\hat{f}_{\mathcal{T},t}(x) = \sum_{n \in \mathcal{L}(\mathcal{T})} \hat{f}_{n,k_n,t}(x), \quad x \in \mathcal{X}.$$

*We denote $\mathcal{P}(\mathcal{T}_0)$ the set of all prunings of $\mathcal{T}_0$.*

Note that a pruning is a decision tree whose predictions are induced by its leaves, contrary to the core tree $\mathcal{T}_0$. In particular, a prediction made by a leaf of a pruning is inherited from the associated node in $\mathcal{T}_0$ before pruning. We provide some illustration in Figure 2.

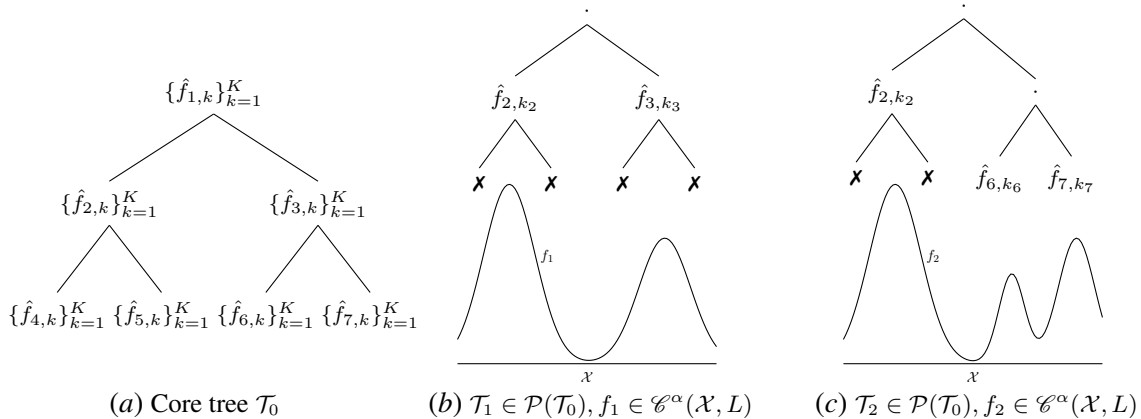

(a) Core tree $\mathcal{T}_0$   (b) $\mathcal{T}_1 \in \mathcal{P}(\mathcal{T}_0), f_1 \in \mathscr{C}^\alpha(\mathcal{X}, L)$   (c) $\mathcal{T}_2 \in \mathcal{P}(\mathcal{T}_0), f_2 \in \mathscr{C}^\alpha(\mathcal{X}, L)$

Figure 2: Example of a core tree $\mathcal{T}_0$ with depth $\mathrm{d}(\mathcal{T}_0) = 3$, $d = 1$, in Fig. 2(a). We give 2 pruned tree instances $\mathcal{T}_1$ for a given Lipschitz function $f_1$ in Fig. 2(b) and $\mathcal{T}_2$ for a second profile $f_2$ in Fig. 2(c). In Fig. 2(a) all nodes $\mathcal{N}(\mathcal{T}_0)$ are awaken and predictive while $\mathcal{T}_1$ in Fig. 2(b) (resp. $\mathcal{T}_2$ in Fig. 2(c)) predicts with $\hat{f}_{2,k_2}, \hat{f}_{3,k_3}$ sitting in its leaves $\mathcal{L}(\mathcal{T}_1)$ (resp. with $\hat{f}_{2,k_2}, \hat{f}_{6,k_6}, \hat{f}_{7,k_7}$ sitting in its leaves $\mathcal{L}(\mathcal{T}_2)$). ✗ represents a pruned node.

**Complexity.** Similar to before, even though our core tree $\mathcal{T}_0$ involves at most $O(|\mathcal{N}(\mathcal{T}_0)|) = O(\sqrt{T} 2^{\mathrm{d}(\mathcal{T}_0)d}) = O(T^{\frac{3}{2}})$ predictors after $T$ iterations, our algorithm remains computationally feasible, since at a time $t$, only a subset of $\mathrm{d}(\mathcal{T}_0)$ nodes are active and updated with the `weight` subroutine. The resulting overall complexity is of order $\frac{1}{d^2} \sqrt{T} \log_2(T)^2$ per step.

**Second result.** In our main result (Theorem 2), we prove that Algorithm 2 achieves a locally adaptive regret with respect to any $\alpha$-Hölder function. Indeed, we show an upper-bound regret that scales with the local regularities of the competitor. Meanwhile, we show that Algorithm 2 also adapts to the curvature of the losses: its regret performances improve when facing exp-concave losses (i.e., when $y \mapsto e^{-\eta \ell_t(y)}$ are concave for some $\eta > 0$), as shown in the second part of Theorem 2. Exp-concave losses include the squared, logistic or logarithmic losses. Note that for Assumption 2 to hold, the gradients $\tilde{\mathbf{g}}_t$ must be bounded by $G$ in the sup-norm. The Hölder assumption on $f$ and the boundedness condition on $\mathcal{X}$ alone are not sufficient. It is also essential that all predictions $\hat{f}_{n,k,t}(x_t)$ are bounded, which is achieved through clipping in Algorithm 2 - see e.g., Gaillard and Gerchinovitz (2015); Cutkosky and Orabona (2018). To simplify the presentation, we state the theorem here only for the case $d = 1$ and $\alpha > 1/2$.

**Theorem 2** *Let $\alpha \in (\frac{1}{2}, 1], d = 1, T \geqslant 1$ and $(\mathcal{T}_0, \bar{\mathcal{X}}, \bar{\mathcal{W}})$ be a core regular tree with $\mathcal{X}_{\mathrm{root}(\mathcal{T}_0)} = \mathcal{X}$ and CT $\{\mathcal{T}_{n,k} : (n, k) \in \mathcal{N}(\mathcal{T}_0) \times [K]\}$ satisfying the same assumptions as in Theorem 1 and whose nodes root are initialized as $\theta_{\mathrm{root}(\mathcal{T}_{n,k}),1} = \gamma_k \in \Gamma$, for all $(n, k) \in \mathcal{N}(\mathcal{T}_0) \times [K]$. Then, Algorithm 2 with a* `weight` *subroutine as in Assumption 2, achieves the regret upper-bound with respect to any $f \in \mathscr{C}^\alpha(\mathcal{X}, L), L > 0$,*

$$\mathrm{Reg}_T(f) \lesssim \inf_{\mathcal{T} \in \mathcal{P}(\mathcal{T}_0)} \left\{ \sqrt{|\mathcal{L}(\mathcal{T})|T} + |\mathcal{L}(\mathcal{T})| + |\mathcal{X}|^\alpha \sum_{n \in \mathcal{L}(\mathcal{T})} L_n(f) 2^{-\alpha(\mathrm{d}(n)-1)} \sqrt{|T_n|} \right\},$$

*where $\lesssim$ is a rough inequality depending on $C_i, i = 1, \ldots, 4, G$ and $L_n(f) \leqslant L, n \in \mathcal{L}(\mathcal{T})$, are the local Hölder constants (7) of $f$, and $T_n = \{1 \leqslant t \leqslant T : x_t \in \mathcal{X}_n\}$.*
*Moreover, if $\ell_1, \ldots, \ell_T$ are exp-concave, one has:*

$$\mathrm{Reg}_T(f) \lesssim \inf_{\mathcal{T} \in \mathcal{P}(\mathcal{T}_0)} \left\{ |\mathcal{L}(\mathcal{T})| + |\mathcal{X}|^\alpha \sum_{n \in \mathcal{L}(\mathcal{T})} L_n(f) 2^{-\alpha(\mathrm{d}(n)-1)} \sqrt{|T_n|} \right\}$$

*where $\lesssim$ also depends on the exp-concavity constant.*

We state and prove a complete version of Theorem 2 in Appendix B, for all $\alpha \in (0, 1], d \geqslant 1$. As a remark, Algorithm 2 is not only adaptive to the local Hölderness of $f$ (via $L_n(f)$), but also to the smoothness rate $\alpha \in (0, 1]$. One could extend the previous results in Theorem 2 with some local smoothness $(\alpha_n)$ associated to the regularity of the function over the pruned leaves at the price of the interpretability of the bound in specific situations as below.

**Minimax optimality and adaptivity to the loss curvature.** Moreover, Theorem 2 yields the following corollary, which demonstrates that our algorithm simultaneously achieves optimal rates for generic convex losses (i.e., similar rates to Theorem 1) and for exp-concave losses, while also adapting locally to the Hölder profile of the competitor - i.e. exhibiting dependencies to constants $L_n(f)$ of the target function $f$. Importantly, our algorithm does not require prior knowledge of the curvature of the losses.

**Corollary 1** *Let $d = 1$ and $\alpha \in (\frac{1}{2}, 1]$. Under assumptions of Theorem 2, Algorithm 2 achieves a regret with respect to any $f \in \mathscr{C}^\alpha(\mathcal{X}, L), L > 0$, and any pruning $\mathcal{T} \in \mathcal{P}(\mathcal{T}_0)$,*

$$\mathrm{Reg}_T(f) \lesssim \inf_{\mathcal{T} \in \mathcal{P}(\mathcal{T}_0)} \left\{ \sum_{n \in \mathcal{L}(\mathcal{T})} \left( L_n(f) |\mathcal{X}_n|^\alpha \right)^{\frac{1}{2\alpha}} \sqrt{|T_n|} \right\}.$$

*Moreover, if $\ell_1, \ldots, \ell_T$ are exp-concave, one has:*

$$\mathrm{Reg}_T(f) \lesssim \inf_{\mathcal{T} \in \mathcal{P}(\mathcal{T}_0)} \left\{ \sum_{n \in \mathcal{L}(\mathcal{T})} \left( L_n(f) |\mathcal{X}_n|^\alpha \right)^{\frac{2}{2\alpha+1}} |T_n|^{\frac{1}{2\alpha+1}} \right\}.$$

The proof of Corollary 1 is postponed to Appendix C. In particular, upper-bounding the infimum over all prunings by the root, our regret becomes $O(L^{2/(2\alpha+1)}T^{1/(2\alpha+1)})$ and $O(L^{1/(2\alpha)}\sqrt{T})$ for the exp-concave and general case respectively. This achieves the same optimal regret to that obtained in Gaillard and Gerchinovitz (2015), for any sequence of exp-concave losses, without the prior-knowledge of the scale-parameter $\gamma$ that they require, and adapting to any regularity while they consider $L, \alpha = 1$. Our algorithm is also nearly minimax in term of the constants $(L, \alpha)$ as shown by Tsybakov (2008), Hazan and Megiddo (2007) or Bach (2024). We provide some experimental illustrations of the results from Corollary 1 in Appendix F.

We note that the fast rate in $T$ obtained under exp-concavity is not optimal in $L$. Thus a compromise is made by our algorithm which competes with more complex oracle trees when $L$ is large to improve and obtain the rate $\sqrt{L}$ by decreasing the rate in $T$. Such trade-off is classical in parametric online learning as bearing resemblances with the comparison between first and second order algorithms, the first ones being optimal in the dimension, the second ones in $T$. Remarkably, our unique algorithm achieves both regret bounds which opens the door to a minimax theory on rates in $L$ and $T$ and not solely on fast rates in $T$.

**Adaptivity to local regularities.** Theorem 2 improves the optimal regret bound established in Theorem 1 by making it adaptive to the local regularities of the Hölder function $f$. To illustrate this better, applying Hölder's inequality entails (see Appendix D for details): for any pruning $\mathcal{T}$

$$
\operatorname{Reg}_T(f) \lesssim \begin{cases} (|\mathcal{X}|^\alpha \bar{L}(f))^{\frac{2}{2\alpha+1}} T^{\frac{1}{2\alpha+1}} & \text{if } \ell_t \text{ are exp-concave}, \\ (|\mathcal{X}|^\alpha \bar{L}(f))^{\frac{1}{2\alpha}} \sqrt{T}, \end{cases} \tag{9}
$$

where $\bar{L}(f) = \left(\frac{1}{|\mathcal{X}|}\sum_{n\in\mathcal{L}(\mathcal{T})}|\mathcal{X}_n|L_n(f)^{1/\alpha}\right)^\alpha$ is an average of the local Hölder constants $L_n(f)$ weighted by the size of the sets $\mathcal{X}_n$ over $\mathcal{T}$. This result is in the same spirit as that of Kuzborskij and Cesa-Bianchi (2020), that focus on adapting to tree-based local Lipschitz profiles. However, contrary to us, they need to assume the prior knowledge of bounds $(M^{(k)})_{1\leqslant k\leqslant\mathrm{d}(\mathcal{T}_0)}$ such that $M^{(k)} \geqslant L_n(f)$ for any $n \in \mathcal{N}(\mathcal{T}_0), \mathrm{d}(n) = k$. Doing so, for any pruning $\mathcal{T}$, when $\alpha = 1$ and ignoring the dependence on $\mathcal{X}$, for the squared loss (which is exp-concave), they prove a bound of order

$$
O\left((\bar{M}(f)T)^{\frac{1}{2}} + \sum_k(M^{(k)}|T^{(k)}|)^{1/2}\right) \qquad \text{where} \qquad \bar{M}(f) = \sum_{k=1}^{\mathrm{d}(\mathcal{T})} w^{(k)}M^{(k)},
$$

with $w^{(k)}$ the proportion of leaves at depth $k$ in the pruning; and $T^{(k)}$ the set of rounds in which $x_t$ belongs to a leaf at level $k$. By grouping our leaves $n$ by their respective depths and applying Hölder's inequality, our results recover theirs with two key improvements (see Appendix E for details): (1) the prior-knowledge of the $M^{(k)}$ is not required in our case and they are replaced with the true local Hölder constants $L_n(f)$ that are smaller; (2) the rate in $T$ is improved from $\sqrt{T}$ to $T^{1/3}$. Note that, similarly to us, the results of Kuzborskij and Cesa-Bianchi (2020) hold for general dimensions and convex losses as well.

## 4. Conclusion and perspectives

In this paper we introduced an online learning approach based on chaining trees and proved that this method achieves minimax regret for the $\alpha$-Hölder nonparametric regression problem, $\alpha \in (0, 1]$. We designed a general and computationally tractable algorithm that leverages a core structure based on chaining-trees to perform an optimal local approximation of $\alpha$-Hölder functions, where $\alpha \leqslant 1$.

In addition, we showed that our algorithm adapts to the curvature of the loss functions revealed by the environment, while remaining optimal in a minimax sense. A limitation of our approach is that chaining trees are minimax-competitive only against $\alpha$-Hölder continuous functions when the smoothness parameter $\alpha \in (0, 1]$. However, combinations of trees, such as the forests studied in Arlot and Genuer (2014) and Mourtada et al. (2020), achieve minimax rates for $\alpha \in (1, 2]$. Since their framework is based on batch i.i.d. data, an open question remains as to whether combinations of chaining trees can also be minimax-optimal in an adversarial setting against functions with higher regularity.

As future work, our approach could be extended to incorporate alternative structures beyond chaining trees, such as kernels or shallow networks. In particular, employing other function approximation methods could address a nonparametric regression problem with respect to richer classes of functions.

**Link with boosting in adversarial online regression.** Boosting is a well-established strategy in statistical learning (Friedman, 2001; Zhang and Yu, 2005), where a set of weak learners is iteratively combined to construct a strong predictor with improved accuracy. Conceptually, this process refines predictions at each step by correcting errors from previous iterations. Our approach shares similarities with boosting-based methods in that it iteratively and adaptively refines function approximations over time. For instance, the structure of chaining trees that we studied can be seen as an implicit hierarchical refinement process, akin to boosting's combination of weak learners. While boosting has been extensively studied in batch settings, recent research (Beygelzimer et al., 2015; Hazan and Singh, 2021) has encouraged the adaptation and study of boosting procedures in the context of adversarial nonparametric regression.

A natural question is whether exposing our algorithms at a meta-state could provide a foundation for analyzing more general weak learners in the context of adversarial online regression. Specifically, instead of relying on a pre-defined hierarchical structure such as chaining trees, one could explore dynamically learning general weak function approximators (e.g., shallow trees, shallow networks) and adaptively aggregating them over time. This perspective is motivated by a more general form of our Algorithm 1, which we expose here.

Figure 3: Boosting at time $t$.

Let $\mathcal{W}$ be a set of real-valued functions $\mathcal{X} \to \mathbb{R}$, and for some $N \geqslant 1$, define the function space:

$$\text{span}_N(\mathcal{W}) = \Big\{ \sum_{n=1}^{N} \beta_n h_n, \ h_n \in \mathcal{W}, \beta_n \in \mathbb{R} \Big\}, \quad (10)$$

which forms a linear space of functions based on $N$ elements from $\mathcal{W}$. The goal is to find a sequence of functions $\hat{f}_t \in \text{span}_N(\mathcal{W})$, for $t \geqslant 1$, such that it minimizes the regret $\text{Reg}_T(\mathcal{F})$ as defined in (1), with $\mathcal{F} = \text{span}_N(\mathcal{W})$.

To illustrate this general perspective, one could present our Algorithm 1 as an abstract formulation of a boosting-like procedure for function approximation based on a gradient update. A schematic diagram is provided in Figure 3. Specifically, at each step $t \geqslant 1$, a meta-version of Algorithm 1 would perform Equation (3) to find a pair $(\beta_{n,t+1}, h_{n,t+1}) \in \mathbb{R} \times \mathcal{W}$ approximating a minimum of the following objective function

$$(\beta_n, h_n) \mapsto \ell_t(\hat{f}_{-n,t}(x_t) + \beta_n h_n(x_t)) \quad \text{where} \quad \hat{f}_{-n,t}(x_t) = \hat{f}_t(x_t) - \beta_{n,t} h_{n,t}(x_t) \quad (11)$$

using the gradient $\left[ \nabla_{(\beta_n, h_n)} \ell_t \big( \hat{f}_{-n,t}(x_t) + \beta_n h_n(x_t) \big) \right]_{(\beta_n, h_n) = (\beta_{n,t}, h_{n,t})}$.

In Section 2, we analyzed the special case where $\mathcal{W}$ is specified as $\{ h_n : x \mapsto \theta_n \mathbb{1}_{x \in \mathcal{X}_n}, \theta_n \in \mathbb{R}, n \in \mathcal{N}(\mathcal{T}) \}$, with fixed $\beta_n = 1$ and $N = |\mathcal{N}(\mathcal{T})|$, using a parameter-free gradient minimization step. A compelling direction for future research is to analyze whether the meta-algorithm defined by Equation (11) can achieve minimax rates under assumptions on weak learners belonging to a general $\mathcal{W}$. By framing the problem in this way, we believe that it could be analyzed more broadly within an online and adversarial boosting framework - see, for instance, Beygelzimer et al. (2015).

## Acknowledgments

We acknowledge the financial and material support of the Sorbonne Center for Artificial Intelligence (SCAI) currently funding the Ph.D scholarship of Paul Liautaud. Pierre Gaillard and Paul Liautaud also thank Institut Pauli CNRS, Wien, for their hospitality during their visits.

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

# APPENDIX

## Appendix A. Proof of Theorem 1

Let $f^* \in \arg\min_{f \in \mathscr{C}^\alpha(\mathcal{X},L)} \sum_{t=1}^T \ell_t(f(x_t))$. We define the function

$$\hat{f}^* = \sum_{n \in \mathcal{L}(\mathcal{T})} f^*(x_n) \mathbb{1}_{\mathcal{X}_n}, \tag{12}$$

where $T_n = \{1 \leqslant t \leqslant T : x_t \in \mathcal{X}_n\}$, $x_n$ the center of hyper region $\mathcal{X}_n$ (i.e. for any $x \in \mathcal{X}_n, \|x - x_n\| \leqslant 2^{-1}|\mathcal{X}_n|$). The proof starts with the following regret decomposition

$$\mathrm{Reg}_T(\mathscr{C}^\alpha(\mathcal{X},L)) = \underbrace{\sum_{t=1}^T \ell_t(\hat{f}_t(x_t)) - \ell_t(\hat{f}^*(x_t))}_{R_1} + \underbrace{\sum_{t=1}^T \ell_t(\hat{f}^*(x_t)) - \ell_t(f^*(x_t))}_{R_2}. \tag{13}$$

We will refer to $R_1$ as the estimation error, which consists of the error incurred by sequentially learning the best Chaining-Tree $\hat{f}^*$. $R_2$ will refer to the approximation error, which involves approximating Hölder functions in $\mathscr{C}^\alpha(\mathcal{X},L)$ by piecewise-constant functions with $|\mathcal{L}(\mathcal{T})|$ pieces.

**Step 1: Upper-bounding the approximation error $R_2$.** Note that by definition of the Chaining-Tree $\mathcal{T}$ (see Definition 1), $\{\mathcal{X}_n, n \in \mathcal{L}(\mathcal{T})\}$ forms a partition of $\mathcal{X} = \mathcal{X}_{\mathrm{root}(\mathcal{T})}$ and for any leaf $n \in \mathcal{L}(\mathcal{T})$

$$|\mathcal{X}_n| = \frac{|\mathcal{X}_{\mathrm{root}(\mathcal{T})}|}{2^{\mathrm{d}(n)-1}} = \frac{|\mathcal{X}|}{2^{\mathrm{d}(\mathcal{T})-1}}. \tag{14}$$

Then,

$$
\begin{aligned}
R_2 &= \sum_{t=1}^T \ell_t(\hat{f}^*(x_t)) - \ell_t(f^*(x_t)) \\
&\leqslant \sum_{t=1}^T G|\hat{f}^*(x_t) - f^*(x_t)| && \leftarrow \ell_t \text{ is } G\text{-Lipschitz} \\
&= G \sum_{t=1}^T \Big| \sum_{n \in \mathcal{L}(\mathcal{T})} f^*(x_n) \mathbb{1}_{x_t \in \mathcal{X}_n} - f^*(x_t) \Big| && \leftarrow \text{by (12)} \\
&= G \sum_{n \in \mathcal{L}(\mathcal{T})} \sum_{t \in T_n} |f^*(x_n) - f^*(x_t)| && \leftarrow \{\mathcal{X}_n, n \in \mathcal{L}(\mathcal{T})\} \text{ partitions } \mathcal{X} \\
&\leqslant G \sum_{n \in \mathcal{L}(\mathcal{T})} \sum_{t \in T_n} L\|x_n - x_t\|_\infty^\alpha && \leftarrow f^* \in \mathscr{C}^\alpha(\mathcal{X},L) \\
&\leqslant G \sum_{n \in \mathcal{L}(\mathcal{T})} L 2^{-\alpha} |\mathcal{X}_n|^\alpha |T_n| && \leftarrow x_n \text{ center of } \mathcal{X}_n \\
&\leqslant GL 2^{-\alpha \mathrm{d}(\mathcal{T})} |\mathcal{X}|^\alpha T, \tag{15}
\end{aligned}
$$

where the last inequality is by (14) and because the leaves form a partition of $\mathcal{X}$, which implies $\sum_{n \in \mathcal{L}(\mathcal{T})} |T_n| = T$.

**Step 2: Upper-bounding the estimation error** $R_1$**.** We now turn to the bound of the estimation error, that is the regret with respect to best Chaining-Tree $\hat{f}^*$.

*Step 2.1: Parametrization of $\hat{f}^*$ in terms of $\theta_n$.* Note that the parametrization of $\hat{f}^*$ in terms of $\theta_n$ is non-unique. We design below a parametrization such that for any $x \in \mathcal{X}$

$$\hat{f}^*(x) = \sum_{n \in \mathcal{N}(\mathcal{T})} \theta_n \mathbb{1}_{x \in \mathcal{X}_n}, \tag{16}$$

and which will allow us to leverage the chaining structure of our Chaining-Tree. We define,

$$\theta_{\text{root}(\mathcal{T})} = f^*(x_{\text{root}(\mathcal{T})}) \quad \text{and} \quad \theta_n = f^*(x_n) - f^*(x_{\text{p}(n)}), \quad \text{for } n \neq \text{root}(\mathcal{T}), \tag{17}$$

where $T_n = \{1 \leqslant t \leqslant T : x_t \in \mathcal{X}_n\}$ and $x_n$ stands for the center of subregion $\mathcal{X}_n$ for any $n \in \mathcal{N}(\mathcal{T})$.

Let us show that the above construction (17) indeed satisfies (16). To do so, we fix $x \in \mathcal{X}$ and proceed by induction on $m = 1, \ldots, \text{d}(\mathcal{T})$, by proving that

$$\sum_{n \in \mathcal{N}(\mathcal{T})} \theta_n \mathbb{1}_{x \in \mathcal{X}_n} \mathbb{1}_{\text{d}(n) \leqslant m} = \sum_{n \in \mathcal{N}(\mathcal{T})} f^*(x_n) \mathbb{1}_{x \in \mathcal{X}_n} \mathbb{1}_{\text{d}(n) = m}. \tag{$\mathcal{H}_m$}$$

First, note that $(\mathcal{H}_1)$ is true by definition of $\theta_{\text{root}(\mathcal{T})}$. Then, let $m \geqslant 1$, and assume that $(\mathcal{H}_m)$ is satisfied, we have

$$\sum_{n \in \mathcal{N}(\mathcal{T})} \theta_n \mathbb{1}_{x \in \mathcal{X}_n} \mathbb{1}_{\text{d}(n) \leqslant m+1}$$

$$= \sum_{n \in \mathcal{N}(\mathcal{T})} \theta_n \mathbb{1}_{x \in \mathcal{X}_n} \mathbb{1}_{\text{d}(n) \leqslant m} + \sum_{n \in \mathcal{N}(\mathcal{T})} \theta_n \mathbb{1}_{x \in \mathcal{X}_n} \mathbb{1}_{\text{d}(n) = m+1}$$

$$= \sum_{n \in \mathcal{N}(\mathcal{T})} f^*(x_n) \mathbb{1}_{x \in \mathcal{X}_n} \mathbb{1}_{\text{d}(n) = m} + \sum_{n \in \mathcal{N}(\mathcal{T})} \theta_n \mathbb{1}_{x \in \mathcal{X}_n} \mathbb{1}_{\text{d}(n) = m+1} \qquad \leftarrow \text{by } (\mathcal{H}_m)$$

$$= \sum_{n \in \mathcal{N}(\mathcal{T})} f^*(x_n) \mathbb{1}_{x \in \mathcal{X}_n} \mathbb{1}_{\text{d}(n) = m}$$

$$\qquad\qquad + \sum_{n \in \mathcal{N}(\mathcal{T})} (f^*(x_n) - f^*(x_{\text{p}(n)})) \mathbb{1}_{x \in \mathcal{X}_n} \mathbb{1}_{\text{d}(n) = m+1} \qquad \leftarrow \text{by } (17)$$

$$= \sum_{n \in \mathcal{N}(\mathcal{T})} f^*(x_n) \mathbb{1}_{x \in \mathcal{X}_n} \mathbb{1}_{\text{d}(n) = m+1},$$

which concludes the induction. In particular, for $m = \text{d}(\mathcal{T})$, $(\mathcal{H}_m)$ yields

$$\sum_{n \in \mathcal{N}(\mathcal{T})} \theta_n \mathbb{1}_{x \in \mathcal{X}_n} = \sum_{n \in \mathcal{L}(\mathcal{T})} f^*(x_n) \mathbb{1}_{x \in \mathcal{X}_n} = \hat{f}^*(x),$$

where the last equality is by definition of $\hat{f}^*$ in (12).

*Step 2.2: Upper-bounding $|\theta_n|$.* The key advantage of the parametrization $\theta_n$ in (17) is that it leverages the chaining structure of our tree. Each node aims to correct the error made by its parent, and as we show below, this error decreases significantly with the depth $\text{d}(n)$ of the node $n$. Let $n \in \mathcal{N}(\mathcal{T}) \setminus \{\text{root}(\mathcal{T})\}$,

$$|\theta_n| = |f^*(x_n) - f^*(x_{\text{p}(n)})| \leqslant L\|x_n - x_{\text{p}(n)}\|_\infty^\alpha = L2^{-\alpha}|\mathcal{X}_n|^\alpha = L|\mathcal{X}|^\alpha 2^{-\alpha \text{d}(n)} \tag{18}$$

where the last equalities are because $\mathcal{X}_n \subset \mathcal{X}_{\mathrm{p}(n)}$ and $|\mathcal{X}_n| = |\mathcal{X}|2^{-(\mathrm{d}(n)-1)}$, from Definition 1. Furthermore, by definition of $\mathscr{C}^\alpha(\mathcal{X}, L)$, $|f^*(x)| \leqslant B$ for any $x \in \mathcal{X}$, hence

$$|\theta_{\mathrm{root}(\mathcal{T})}| = |f^*(x_{\mathrm{root}(\mathcal{T})})| \leqslant B \,.$$

*Step 2.3: Proof of the regret upper bound.* We are now ready to upper bound the estimation error in (13). We have

$$
\begin{aligned}
R_1 &= \sum_{t=1}^{T} \ell_t(\hat{f}_t(x_t)) - \ell_t(\hat{f}^*(x_t)) \\
&= \sum_{t=1}^{T} \ell_t\big(\textstyle\sum_{n\in\mathcal{N}(\mathcal{T})} \theta_{n,t}\mathbb{1}_{x_t\in\mathcal{X}_n}\big) - \ell_t\big(\textstyle\sum_{n\in\mathcal{N}(\mathcal{T})} \theta_n\mathbb{1}_{x_t\in\mathcal{X}_n}\big) \\
&\leqslant \sum_{t=1}^{T} \sum_{n\in\mathcal{N}(\mathcal{T})} g_{n,t}(\theta_{n,t} - \theta_n)
\end{aligned}
\tag{19}
$$

by convexity of $\ell_t$, where $g_{n,t}$ is the partial subgradient in $\theta_{n,t}$ as defined in Equation (4). Now, from Assumption 1 on the `grad-step` procedure to optimize $\theta_{n,t}$ and with $\theta_{n,1} = 0, g_{n,t} \leqslant G\mathbb{1}_{x_t\in\mathcal{X}_n}$, we further have, with $T_n = \{1 \leqslant t \leqslant T : g_{n,t} \neq 0\}$,

$$
\begin{aligned}
R_1 &\leqslant G \sum_{n\in\mathcal{N}(\mathcal{T})} |\theta_n|(C_1\sqrt{|T_n|} + C_2) \\
&= G \sum_{m=1}^{\mathrm{d}(\mathcal{T})} \sum_{n:\mathrm{d}(n)=m} |\theta_n|(C_1\sqrt{|T_n|} + C_2) \\
&\leqslant BG(C_1\sqrt{T} + C_2) + LG|\mathcal{X}|^\alpha \sum_{m=2}^{\mathrm{d}(\mathcal{T})} \sum_{n:\mathrm{d}(n)=m} (C_1\sqrt{|T_n|} + C_2)2^{-\alpha m} \qquad \leftarrow \text{by (18)}
\end{aligned}
\tag{20}
$$

Now, because in a $d$-regular decision tree, the number of nodes with depth $m$ equals $|\{n : \mathrm{d}(n) = m\}| = 2^{d(m-1)}$ (recall that the depth of the root is 1), and because $\{\mathcal{X}_n : \mathrm{d}(n) = m\}$ forms a partition of $\mathcal{X}$, we have $\sum_{n:\mathrm{d}(n)=m} T_n = T$ and by Cauchy-Schwarz inequality

$$\sum_{n:\mathrm{d}(n)=m} \sqrt{T_n} \leqslant \sqrt{2^{d(m-1)}\textstyle\sum_{n:\mathrm{d}(n)=m}T_n} = \sqrt{2^{d(m-1)}T} \,,$$

which substituted into the previous upper bound entails

$$
\begin{aligned}
R_1 &\leqslant BG(C_1\sqrt{T} + C_2) + LG|\mathcal{X}|^\alpha \sum_{m=2}^{\mathrm{d}(\mathcal{T})} \Big(C_1 2^{\frac{d(m-1)}{2}-\alpha m}\sqrt{T} + C_2 2^{d(m-1)-\alpha m}\Big) \\
&= BG(C_1\sqrt{T} + C_2) + LG|\mathcal{X}|^\alpha \Big(2^{-\frac{d}{2}}C_1\sqrt{T} \sum_{m=2}^{\mathrm{d}(\mathcal{T})} 2^{m(\frac{d}{2}-\alpha)} + 2^{-d}C_2 \sum_{m=2}^{\mathrm{d}(\mathcal{T})} 2^{m(d-\alpha)}\Big).
\end{aligned}
\tag{21}
$$

**Step 3: Conclusion and optimization of** $d(\mathcal{T})$**.** To conclude the proof, we consider three cases according to the sign of $d - 2\alpha$:

• *Case 1: if $d < 2\alpha$.* Then

$$2^{-\frac{d}{2}} \sum_{m=2}^{d(\mathcal{T})} 2^{m(\frac{d}{2}-\alpha)} \leqslant \frac{1}{1 - 2^{\frac{d}{2}-\alpha}} \quad \text{and} \quad 2^{-d} \sum_{m=2}^{d(\mathcal{T})} 2^{m(d-\alpha)} \leqslant 2^{-d} \sum_{m=0}^{d(\mathcal{T})} 2^{m\alpha} \leqslant 2^{-d} \frac{2^{\alpha(d(\mathcal{T})+1)}}{2^{\alpha}-1} \overset{(2\alpha \geqslant 1)}{\leqslant} 2^{\alpha d(\mathcal{T})+2} ,$$

and (21) yields

$$R_1 \leqslant BG(C_1\sqrt{T} + C_2) + LG|\mathcal{X}|^{\alpha}\left(\frac{C_1\sqrt{T}}{1 - 2^{\frac{d}{2}-\alpha}} + C_2 2^{\alpha d(\mathcal{T})+2}\right);$$

Therefore, combining with (13) and (15), the regret is upper-bounded as

$$\mathrm{Reg}_T(\mathscr{C}^{\alpha}(\mathcal{X}, L)) \leqslant BG(C_1\sqrt{T} + C_2) + LG|\mathcal{X}|^{\alpha}\left(\frac{C_1\sqrt{T}}{1 - 2^{\frac{d}{2}-\alpha}} + C_2 2^{\alpha d(\mathcal{T})+2} + T2^{-\alpha d(\mathcal{T})}\right).$$

The choice $d(\mathcal{T}) = \frac{1}{d}\log_2 T$ entails

$$\mathrm{Reg}_T(\mathscr{C}^{\alpha}(\mathcal{X}, L)) \leqslant BG(C_1\sqrt{T} + C_2) + LG|\mathcal{X}|^{\alpha}\left(\frac{C_1}{1 - 2^{\frac{d}{2}-\alpha}} + 4C_2 + 1\right)\sqrt{T}. \qquad (22)$$

• *Case 2: if $d = 2\alpha$.* Then

$$2^{-\frac{d}{2}} \sum_{m=2}^{d(\mathcal{T})} 2^{m(\frac{d}{2}-\alpha)} \leqslant d(\mathcal{T}) \quad \text{and} \quad 2^{-d} \sum_{m=2}^{d(\mathcal{T})} 2^{m(d-\alpha)} = 2^{-d} \sum_{m=2}^{d(\mathcal{T})} 2^{m\alpha} \leqslant 2^{\alpha d(\mathcal{T})+2} ,$$

and (21) yields

$$R_1 \leqslant BG(C_1\sqrt{T} + C_2) + LG|\mathcal{X}|^{\alpha}\left(C_1\sqrt{T}d(\mathcal{T}) + C_2 2^{\alpha d(\mathcal{T})+2}\right);$$

Therefore, combining with (13) and (15), the regret is upper-bounded as

$$\mathrm{Reg}_T(\mathscr{C}^{\alpha}(\mathcal{X}, L)) \leqslant BG(C_1\sqrt{T} + C_2) + 2^{\alpha}LG|\mathcal{X}|^{\alpha}\left(C_1\sqrt{T}d(\mathcal{T}) + C_2 2^{\alpha d(\mathcal{T})+2} + T2^{-\alpha d(\mathcal{T})}\right).$$

The choice $d(\mathcal{T}) = \frac{1}{d}\log_2 T$ entails

$$\mathrm{Reg}_T(\mathscr{C}^{\alpha}(\mathcal{X}, L)) \leqslant BG(C_1\sqrt{T} + C_2) + 2^{\alpha}LG|\mathcal{X}|^{\alpha}\left(\frac{C_1}{d}\log_2 T + 4C_2 + 1\right)\sqrt{T}. \qquad (23)$$

• *Case 3: if $d > 2\alpha$.* Then

$$2^{-\frac{d}{2}} \sum_{m=2}^{d(\mathcal{T})} 2^{m(\frac{d}{2}-\alpha)} \leqslant \frac{2^{(\frac{d}{2}-\alpha)d(\mathcal{T})}}{2^{\frac{d}{2}-\alpha}-1} \quad \text{and} \quad 2^{-d} \sum_{m=2}^{d(\mathcal{T})} 2^{m(d-\alpha)} \leqslant \frac{2^{(d-\alpha)d(\mathcal{T})}}{2^{d-\alpha}-1} \leqslant 2^{(d-\alpha)d(\mathcal{T})+2} ,$$

where the last inequality is because $(2^{d-\alpha} - 1)^{-1} \leqslant (2^{d/2} - 1)^{-1} \leqslant (\sqrt{2} - 1)^{-1} \leqslant 4$. And (21) yields

$$R_1 \leqslant BG(C_1\sqrt{T} + C_2) + LG|\mathcal{X}|^{\alpha}\left(C_1\sqrt{T}\frac{2^{(\frac{d}{2}-\alpha)d(\mathcal{T})}}{2^{\frac{d}{2}-\alpha}-1} + C_2 2^{(d-\alpha)d(\mathcal{T})+2}\right).$$

Therefore, combining with (13) and (15), the regret is upper-bounded as

$$\mathrm{Reg}_T(\mathscr{C}^\alpha(\mathcal{X}, L)) \leqslant BG(C_1\sqrt{T}+C_2)+LG|\mathcal{X}|^\alpha\left(C_1\sqrt{T}\frac{2^{(\frac{d}{2}-\alpha)\mathrm{d}(\mathcal{T})}}{2^{\frac{d}{2}-\alpha}-1}+C_2 2^{(d-\alpha)\mathrm{d}(\mathcal{T})+2}+T2^{-\alpha\mathrm{d}(\mathcal{T})}\right).$$

The choice $\mathrm{d}(\mathcal{T}) = \frac{1}{d}\log_2 T$ entails

$$\mathrm{Reg}_T(\mathscr{C}^\alpha(\mathcal{X}, L)) \leqslant BG(C_1\sqrt{T}+C_2) + LG|\mathcal{X}|^\alpha\left(\frac{C_1}{2^{\frac{d}{2}-\alpha}-1}+4C_2+1\right)T^{1-\frac{\alpha}{d}}. \qquad (24)$$

*Conclusion.* Combining the three cases (22), (23), and (24) concludes the proof of the regret bound, which we summarize below

$$\mathrm{Reg}_T(\mathscr{C}^\alpha(\mathcal{X}, L)) \leqslant BG(C_1\sqrt{T}+C_2)+GL|\mathcal{X}|^\alpha\begin{cases}\left(\Phi(\frac{d}{2}-\alpha)C_1+4C_2+1\right)\sqrt{T} & \text{if } d < 2\alpha \\ \left(\frac{C_1}{d}\log_2 T+4C_2+1\right)\sqrt{T} & \text{if } d = 2\alpha \\ \left(\Phi(\frac{d}{2}-\alpha)C_1+4C_2+1\right)T^{1-\frac{\alpha}{d}} & \text{if } d > 2\alpha,\end{cases}$$

where $\Phi(u) = |2^u - 1|^{-1}$.

## Appendix B. Proof of Theorem 2

We state here the full version of Theorem 2 that we prove right after.

**Theorem 3** *Let $T, d \geqslant 1$ and $(\mathcal{T}_0, \bar{\mathcal{X}}, \bar{\mathcal{W}})$ be a core regular tree with CT $\{\mathcal{T}_{n,k}, (n,k) \in \mathcal{N}(\mathcal{T}_0) \times [K]\}$ satisfying the same assumptions as in Theorem 1 and root nodes initialized as $\theta_{\mathrm{root}(\mathcal{T}_{n,k}),1} = \gamma_k \in \Gamma$, for all $(n,k) \in \mathcal{N}(\mathcal{T}_0) \times [K]$. Then, Algorithm 2 with a* `weight` *subroutine as in Assumption 2, achieves the regret upper-bound with respect to any $f \in \mathscr{C}^\alpha(\mathcal{X}, L), L > 0$,*

$$
\mathrm{Reg}_T(f) \leqslant \inf_{\mathcal{T} \in \mathcal{P}(\mathcal{T}_0)} \left\{ \beta_1 \sqrt{T|\mathcal{L}(\mathcal{T})|} + \beta_2 |\mathcal{L}(\mathcal{T})| \right.
$$

$$
\left. + G|\mathcal{X}|^\alpha \sum_{n \in \mathcal{L}(\mathcal{T})} L_n(f) 2^{-\alpha(\mathrm{d}(n)-1)} \begin{cases} \psi_1 \sqrt{|T_n|} & \text{if } d < 2\alpha \\ \psi_2 \log_2 |T_n| \sqrt{|T_n|} & \text{if } d = 2\alpha \\ \psi_1 |T_n|^{1-\frac{\alpha}{d}} & \text{if } d > 2\alpha, \end{cases} \right\},
$$

*with $\beta_1 = 2C_3 G \sqrt{\log\left(2BT|\mathcal{N}(\mathcal{T}_0)|\right)}$ and $\beta_2 = G(2^{-1}C_1 + C_2 2^{-1} T^{-\frac{1}{2}} + C_4)$, local Lipschitz constants $L_n(f) \leqslant L$ as in (7), $\psi_1 = \Phi(d/2 - \alpha)C_1 + 4C_2 + 1, \psi_2 = C_1/d + 4C_2 + 1$, and $\Phi, C_1, C_2$ as in Theorem 1.*
*Moreover, if $\ell_1, \ldots, \ell_T$ are $\eta$-exp-concave with some $\eta > 0$, one has:*

$$
\mathrm{Reg}_T(f) \leqslant \inf_{\mathcal{T} \in \mathcal{P}(\mathcal{T}_0)} \left\{ \beta_3 |\mathcal{L}(\mathcal{T})| + G|\mathcal{X}|^\alpha \sum_{n \in \mathcal{L}(\mathcal{T})} L_n(f) 2^{-\alpha(\mathrm{d}(n)-1)} \begin{cases} \psi_1 \sqrt{|T_n|} & \text{if } d < 2\alpha \\ \psi_2 \log_2 |T_n| \sqrt{|T_n|} & \text{if } d = 2\alpha \\ \psi_1 |T_n|^{1-\frac{\alpha}{d}} & \text{if } d > 2\alpha, \end{cases} \right\}
$$

*with $\beta_3 = \dfrac{C_3^2 \log\left(2BT|\mathcal{N}(\mathcal{T}_0)|\right)}{2\mu} + C_4 G + 2^{-1} G(C_1 + C_2 T^{-\frac{1}{2}})$ and $0 < \mu \leqslant \min\{1/G, \eta\}$.*

**Proof [of Theorem 3]** Let $L > 0, \alpha \in (0,1], f^* \in \mathscr{C}^\alpha(\mathcal{X}, L)$ and $\varepsilon > 0$ be the precision of the grid $\Gamma$, $K = \lfloor 2B/\varepsilon \rfloor$ the number of experts in each node in $\mathcal{N}(\mathcal{T}_0)$. Let $\mathcal{T} \in \mathcal{P}(\mathcal{T}_0)$ be some pruned tree from $(\mathcal{T}_0, \bar{\mathcal{X}}, \bar{\mathcal{W}})$ with prediction functions $\bar{\mathcal{W}} = \{(\hat{f}_{n,k})_{k \in [K]}, n \in \mathcal{N}(\mathcal{T}_0)\}$ on subsets $\bar{\mathcal{X}} = \{\mathcal{X}_n, n \in \mathcal{N}(\mathcal{T}_0)\}$. We call $\hat{f}_{\mathcal{T}}$ the associated prediction function of pruning $\mathcal{T}$ (see Definition 2) such that at any time $t \geqslant 1$,

$$
\hat{f}_{\mathcal{T},t}(x) = \sum_{n \in \mathcal{L}(\mathcal{T})} \hat{f}_{n,k_n,t}(x), \qquad x \in \mathcal{X},
$$

with $k_n = \arg\min_{k \in [K]} |(-B + (k-1)\varepsilon) - f^*(x_n)|$ the best approximating constant of $f^*(x_n)$ where $x_n \in \mathcal{X}_n$ is the center of the sub-region $\mathcal{X}_n$, i.e. for any $x \in \mathcal{X}_n, \|x - x_n\|_\infty \leqslant 2^{-1}|\mathcal{X}_n|$. We have a decomposition of regret as:

$$
\mathrm{Reg}_T(f) = \underbrace{\sum_{t=1}^T \ell_t(\hat{f}_t(x_t)) - \ell_t(\hat{f}_{\mathcal{T},t}(x_t))}_{=:R_1} + \underbrace{\sum_{t=1}^T \ell_t(\hat{f}_{\mathcal{T},t}(x_t)) - \ell_t(f^*(x_t))}_{=:R_2}, \tag{25}
$$

$R_1$ is the regret related to the estimation error of the core expert tree $\mathcal{T}_0$ compared to some pruning $\mathcal{T}$ from it. On the other hand, $R_2$ is related to the error of the pruning tree $\mathcal{T}$ against some function $f^*$.

**Step 1: Upper-bounding $R_2$ as local chaining tree regrets.** Recall that according to Definition 2, pruning subsets $\{\mathcal{X}_n, n \in \mathcal{L}(\mathcal{T})\}$ form a partition of $\mathcal{X} = \mathcal{X}_{\text{root}(\mathcal{T}_0)}$. Hence, for any $x_t \in \mathcal{X}$, prediction from pruning $\mathcal{T}$ at time $t$ is $\hat{f}_{\mathcal{T},t}(x_t) = \hat{f}_{n,k_n,t}(x_t)$ with $n \in \mathcal{L}(\mathcal{T})$ the unique leaf such that $x_t \in \mathcal{X}_n$ at time $t$. Then, $R_2$ can be written as follows:

$$R_2 = \sum_{t=1}^{T} \sum_{n \in \mathcal{L}(\mathcal{T})} (\ell_t(\hat{f}_{\mathcal{T},t}(x_t)) - \ell_t(f^*(x_t)))\mathbb{1}_{x_t \in \mathcal{X}_n}$$

$$= \sum_{n \in \mathcal{L}(\mathcal{T})} \sum_{t \in T_n} \ell_t(\hat{f}_{n,k_n,t}(x_t)) - \ell_t(f^*(x_t))$$

$$\leqslant \sum_{n \in \mathcal{L}(\mathcal{T})} \sum_{t \in T_n} \ell_t(\tilde{f}_{n,k_n,t}(x_t)) - \ell_t(f^*(x_t)), \tag{26}$$

where we set $T_n = \{1 \leqslant t \leqslant T : x_t \in \mathcal{X}_n\}, n \in \mathcal{L}(\mathcal{T})$ and (26) is because $\hat{f}_{n,k_n,t} = [\tilde{f}_{n,k_n,t}]_B \leqslant \tilde{f}_{n,k_n,t}$ and $\ell_t$ is convex and has minimum in $[-B, B]$.

The decomposition in (26) represents a sum of *local* error approximations of the function $f^*$ over the partition $\{\mathcal{X}_n, n \in \mathcal{L}(\mathcal{T})\}$, using predictors $\tilde{f}_{n,k_n}$ located at the leaves of the pruned tree $\mathcal{T}$. Recall that for every $n \in \mathcal{N}(\mathcal{T}_0)$, $\tilde{f}_{n,k_n}$ is a prediction function associated with a CT $\mathcal{T}_{n,k_n}$, where the root node starts from $\theta_{\text{root}(\mathcal{T}_{n,k_n}),1} = -B + (k_n - 1)\varepsilon \in \Gamma$ on $\mathcal{X}_n$. In proof of Theorem 1 (Appendix A) we study a regret bound (13) decomposed into two terms: estimation and approximation. In particular, we showed that any CT adapts to any regularity $(L, \alpha) \in \mathbb{R}_+ \times (0, 1]$ of $f^*$. Thus, the approximation error of CT $\tilde{f}_{n,k_n}$ with respect to $f^*$ remains similar to that in (15), but now with regard to an Hölder function with a constant $L_n(f^*) \geqslant 0$ over $\mathcal{X}_n$. Specifically, from (26), we get:

$$R_2 \leqslant \sum_{n \in \mathcal{L}(\mathcal{T})} \left[ G \underbrace{\sum_{m=1}^{\text{d}(\mathcal{T}_{n,k_n})} \sum_{n':\text{d}(n')=m} |\theta_{n'} - \theta_{n',1}|(C_1\sqrt{|T_{n'}|} + C_2)}_{\text{estimation error as in (20)}} \right.$$

$$\left. + \underbrace{GL_n(f^*)|\mathcal{X}_n|^\alpha |T_n| 2^{-\alpha(\text{d}(\mathcal{T}_{n,k_n}))}}_{\text{approximation error (15) over } \mathcal{X}_n} \right], \tag{27}$$

with $C_1, C_2$ as in Assumption 1 and where we set in (17),

$$\theta_{\text{root}(\mathcal{T}_{n,k_n})} = f^*(x_{\text{root}(\mathcal{T}_{n,k_n})}) \quad \text{and} \quad \theta_{n'} = f^*(x_{n'}) - f^*(x_{\text{p}(n')}), \quad n' \in \mathcal{N}(\mathcal{T}_{n,k_n}) \backslash \{\text{root}(\mathcal{T}_{n,k_n})\}.$$

In particular, we have for $n' = \text{root}(\mathcal{T}_{n,k_n})$,

$$|\theta_{n'} - \theta_{n',1}| = \left| f^*(x_{\text{root}(\mathcal{T}_{n,k_n})}) - (-B + (k_n - 1))\varepsilon \right| \leqslant \frac{\varepsilon}{2}, \tag{28}$$

by definition of $k_n$ and since $\Gamma = \{-B + (k-1)\varepsilon\}_{k \in [K]}$ is an $\varepsilon$-discretization of the $y$-axis. Moreover, if $n' \in \mathcal{N}(\mathcal{T}_{n,k_n}), \text{d}(n') \geqslant 2$, one has $\theta_{n',1} = 0$ and

$$|\theta_{n'} - \theta_{n',1}| = |\theta_{n'}| \leqslant L_n(f^*)|\mathcal{X}_n|^\alpha 2^{-\alpha\text{d}(n')}, \tag{29}$$

according to (18) with $f^* \in \mathscr{C}^\alpha(\mathcal{X}_n, L_n)$.

Then, following the same optimization steps as for Theorem 1, in each $\mathrm{d}(\mathcal{T}_{n,k_n}), n \in \mathcal{L}(\mathcal{T})$ of (27), we get:

$$R_2 \leqslant G \sum_{n \in \mathcal{L}(\mathcal{T})} \frac{\varepsilon}{2}(C_1\sqrt{|T_n|} + C_2)$$

$$+ G|\mathcal{X}|^\alpha \sum_{n \in \mathcal{L}(\mathcal{T})} L_n(f^*) 2^{-\alpha(\mathrm{d}(n)-1)} \begin{cases} \psi_1\sqrt{|T_n|} & \text{if } d < 2\alpha \\ \psi_2 \log_2|T_n|\sqrt{|T_n|} & \text{if } d = 2\alpha \\ \psi_1|T_n|^{1-\frac{\alpha}{d}} & \text{if } d > 2\alpha \end{cases}$$

with $\psi_1 = \Phi(d/2 - \alpha)C_1 + 4C_2 + 1, \psi_2 = C_1/d + 4C_2 + 1$, and $\Phi$ defined in Theorem 1. Cauchy-Schwarz inequality gives

$$\sum_{n \in \mathcal{L}(\mathcal{T})} (C_1\sqrt{|T_n|} + C_2) \leqslant C_1\sqrt{|\mathcal{L}(\mathcal{T})||T} + C_2|\mathcal{L}(\mathcal{T})|$$

Finally,

$$R_2 \leqslant \frac{\varepsilon}{2}G\left(C_1\sqrt{|\mathcal{L}(\mathcal{T})||T} + C_2|\mathcal{L}(\mathcal{T})|\right)$$

$$+ G|\mathcal{X}|^\alpha \sum_{n \in \mathcal{L}(\mathcal{T})} L_n(f^*) 2^{-\alpha(\mathrm{d}(n)-1)} \begin{cases} \psi_1\sqrt{|T_n|} & \text{if } d < 2\alpha \\ \psi_2 \log_2|T_n|\sqrt{|T_n|} & \text{if } d = 2\alpha \quad (30) \\ \psi_1|T_n|^{1-\frac{\alpha}{d}} & \text{if } d > 2\alpha \end{cases}$$

**Step 2: Upper-bounding the pruning estimation error $R_1$.** We aim at bounding the estimation error $R_1$ due to the error incurred by sequentially learning the best pruned tree prediction and the best root node in $\Gamma$ inside each pruned leaves. Note that at each time $t$, only a subset of nodes of $\mathcal{T}_0$ are active and output predictions: for any time $t \geqslant 1$, let us denote $\mathcal{N}_t \subset \mathcal{N}(\mathcal{T}_0)$ the set of active nodes (i.e. making a prediction) at time $t$. Remark that

$$\hat{f}_t(x_t) = \sum_{n \in \mathcal{N}(\mathcal{T}_0)} \sum_{k=1}^K w_{n,k,t}\hat{f}_{n,k,t}(x_t) \tag{31}$$

$$= \sum_{n \in \mathcal{N}_t} \sum_{k=1}^K \tilde{w}_{n,k,t}\hat{f}_{n',k',t}(x_t) + \sum_{n \notin \mathcal{N}_t} \sum_{k=1}^K \tilde{w}_{n,k,t}\hat{f}_t(x_t), \tag{32}$$

by definition of $\hat{f}_t$ and the so called trick of prediction with sleeping experts, e.g. in Gaillard et al. (2014). Recall that $\tilde{\mathbf{g}}_t = \nabla_{\tilde{\mathbf{w}}_t}\ell_t\big(\sum_{n \in \mathcal{N}_t}\sum_{k=1}^K \tilde{w}_{n,k,t}\hat{f}_{n,k,t}(x_t) + \sum_{n \notin \mathcal{N}_t}\sum_{k=1}^K \tilde{w}_{n,k,t}\hat{f}_t(x_t)\big) \in \mathbb{R}^{|\mathcal{N}(\mathcal{T}_0)| \times K}$, for all $t \geqslant 1$. Then, for all $n \in \mathcal{N}(\mathcal{T}_0), k \in [K]$,

$$\tilde{g}_{n,k,t} = \begin{cases} \ell'_t(\hat{f}_t(x_t))\hat{f}_{n,k,t}(x_t) & \text{if } n \in \mathcal{N}_t, \\ \ell'_t(\hat{f}_t(x_t))\hat{f}_t(x_t) & \text{if } n \notin \mathcal{N}_t. \end{cases} \tag{33}$$

For any $t \geqslant 1$, $n \in \mathcal{L}(\mathcal{T})$ and $k \in [K]$, one has:

$$
\tilde{\mathbf{g}}_t^\top \mathbf{w}_t - \tilde{g}_{n,k,t} = \sum_{n' \in \mathcal{N}(\mathcal{T}_0)} \sum_{k'=1}^{K} w_{n',k',t} \tilde{g}_{n',k',t} - \tilde{g}_{n,k,t}
$$

$$
= \ell_t'(\hat{f}_t(x_t)) \underbrace{\sum_{n' \in \mathcal{N}_t} \sum_{k'=1}^{K} w_{n',k',t} \hat{f}_{n',k',t}(x_t)}_{=\hat{f}_t(x_t)} - \tilde{g}_{n,k,t}
$$

$$
= \ell_t'(\hat{f}_t(x_t)) \Big( \sum_{n' \in \mathcal{N}_t} \sum_{k'=1}^{K} \tilde{w}_{n',k',t} \hat{f}_{n',k',t}(x_t) + \sum_{n' \notin \mathcal{N}_t} \sum_{k'=1}^{K} \tilde{w}_{n',k',t} \hat{f}_t(x_t) \Big) - \tilde{g}_{n,k,t}
$$

$$
= \ell_t'(\hat{f}_t(x_t))
$$

$$
\times \begin{cases} (\hat{f}_t(x_t) - \hat{f}_t(x_t)) \text{ if } n \notin \mathcal{N}_t, \\ \big( \sum_{n' \in \mathcal{N}_t} \sum_{k'=1}^{K} \tilde{w}_{n',k',t} \hat{f}_{n',k',t}(x_t) + \sum_{n' \notin \mathcal{N}_t} \sum_{k'=1}^{K} \tilde{w}_{n',k',t} \hat{f}_t(x_t) - \hat{f}_{n,k,t}(x_t) \big) \text{ else}, \end{cases}
$$

$$
= \begin{cases} 0 & \text{if } n \notin \mathcal{N}_t, \\ \tilde{\mathbf{g}}_t^\top \tilde{\mathbf{w}}_t - \tilde{g}_{n,k,t} & \text{else}, \end{cases}
$$

$$
= (\tilde{\mathbf{g}}_t^\top \tilde{\mathbf{w}}_t - \tilde{g}_{n,k,t}) \mathbb{1}_{x_t \in \mathcal{X}_n}, \tag{34}
$$

where the second equality follows from (31), the third from (32), and the fourth from (33). Finally, we obtain

$$
(\ell_t(\hat{f}_t(x_t)) - \ell_t(\hat{f}_{n,k,t}(x_t))) \mathbb{1}_{x_t \in \mathcal{X}_n} \leqslant \ell_t'(\hat{f}_t(x_t))(\hat{f}_t(x_t) - \hat{f}_{n,k,t}(x_t)) \mathbb{1}_{x_t \in \mathcal{X}_n} \quad \leftarrow \text{by convexity of } \ell_t
$$

$$
= (\tilde{\mathbf{g}}_t^\top \tilde{\mathbf{w}}_t - \tilde{g}_{n,k,t}) \mathbb{1}_{x_t \in \mathcal{X}_n}
$$

$$
= \tilde{\mathbf{g}}_t^\top \mathbf{w}_t - \tilde{g}_{n,k,t} \quad \leftarrow \text{by (34),} \tag{35}
$$

and setting $T_n = \{1 \leqslant t \leqslant T : x_t \in \mathcal{X}_n\}$, $n \in \mathcal{L}(\mathcal{T})$:

$$
R_1 = \sum_{t=1}^{T} \sum_{n \in \mathcal{L}(\mathcal{T})} (\ell_t(\hat{f}_t(x_t)) - \ell_t(\hat{f}_{n,k_n,t}(x_t))) \mathbb{1}_{x_t \in \mathcal{X}_n} \quad \leftarrow \{\mathcal{X}_n, n \in \mathcal{L}(\mathcal{T})\} \text{ partition of } \mathcal{X}
$$

$$
\leqslant \sum_{n \in \mathcal{L}(\mathcal{T})} \sum_{t=1}^{T} (\tilde{\mathbf{g}}_t^\top \mathbf{w}_t - \tilde{g}_{n,k_n,t}) \quad \leftarrow \text{by (35)}
$$

$$
\leqslant \sum_{n \in \mathcal{L}(\mathcal{T})} \Big( C_3 \sqrt{\log(K|\mathcal{N}(\mathcal{T}_0)|)} \sqrt{\sum_{t=1}^{T} (\tilde{\mathbf{g}}_t^\top \mathbf{w}_t - \tilde{g}_{n,k_n,t})^2} + C_4 G \Big) \quad \leftarrow \text{by Assumption 2}
$$

$$
= C_4 G |\mathcal{L}(\mathcal{T})| + C_3 \sqrt{\log(K|\mathcal{N}(\mathcal{T}_0)|)} \sum_{n \in \mathcal{L}(\mathcal{T})} \sqrt{\sum_{t \in T_n} (\tilde{\mathbf{g}}_t^\top \mathbf{w}_t - \tilde{g}_{n,k_n,t})^2}, \tag{36}
$$

where last equality holds because for any $n \in \mathcal{L}(\mathcal{T})$, $\tilde{\mathbf{g}}_t^\top \mathbf{w}_t - \tilde{g}_{n,k_n,t} = 0$ if $x_t \notin \mathcal{X}_n$.

- *Case 1:* $(\ell_t)_{1 \leqslant t \leqslant T}$ *convex.*

Since $\|\tilde{\mathbf{g}}_t\|_\infty \leqslant G, \|\mathbf{w}_t\|_\infty \leqslant 1, t \in [T]$ by Assumption 2 and using Cauchy-Schwartz inequality we get from Equation (36):

$$
\begin{aligned}
R_1 &\leqslant C_4 G |\mathcal{L}(\mathcal{T})| + 2C_3 \sqrt{\log\left(K|\mathcal{N}(\mathcal{T}_0)|\right)} G \sum_{n \in \mathcal{L}(\mathcal{T})} \sqrt{|T_n|} \\
&\leqslant C_4 G |\mathcal{L}(\mathcal{T})| + 2C_3 G \sqrt{\log\left(K|\mathcal{N}(\mathcal{T}_0)|\right)|\mathcal{L}(\mathcal{T})| \sum_{n \in \mathcal{L}(\mathcal{T})} |T_n|} \\
&= C_4 G |\mathcal{L}(\mathcal{T})| + 2C_3 G \sqrt{\log\left(K|\mathcal{N}(\mathcal{T}_0)|\right)|\mathcal{L}(\mathcal{T})|T}.
\end{aligned} \tag{37}
$$

In case of convex losses, we finally have by (25), (30) and (37) :

$$
\operatorname{Reg}_T(f) \leqslant 2C_3 G \sqrt{\log\left(K|\mathcal{N}(\mathcal{T}_0)|\right)|\mathcal{L}(\mathcal{T})|T} + \left(C_2 \frac{\varepsilon}{2} + C_4\right) G |\mathcal{L}(\mathcal{T})| + \frac{\varepsilon}{2} GC_1 \sqrt{|\mathcal{L}(\mathcal{T})|T}
$$

$$
+ G|\mathcal{X}|^\alpha \sum_{n \in \mathcal{L}(\mathcal{T})} L_n(f) 2^{-\alpha(\mathrm{d}(n)-1)} \begin{cases} \psi_1 \sqrt{|T_n|} & \text{if } d < 2\alpha, \\ \psi_2 \log_2 |T_n| \sqrt{|T_n|} & \text{if } d = 2\alpha, \\ \psi_1 |T_n|^{1-\frac{\alpha}{d}} & \text{if } d > 2\alpha, \end{cases}
$$

with $\psi_1, \psi_2$ defined in (30). Taking $\varepsilon = T^{-\frac{1}{2}}, K = \lfloor 2BT^{\frac{1}{2}} \rfloor \leqslant 2BT$, we get:

$$
\operatorname{Reg}_T(f) \leqslant 2C_3 G \sqrt{\log\left(2BT|\mathcal{N}(\mathcal{T}_0)|\right)} \sqrt{|\mathcal{L}(\mathcal{T})|T} + (2^{-1}C_1 + C_2 2^{-1} T^{-\frac{1}{2}} + C_4) G |\mathcal{L}(\mathcal{T})|
$$

$$
+ G|\mathcal{X}|^\alpha \sum_{n \in \mathcal{L}(\mathcal{T})} L_n(f) 2^{-\alpha(\mathrm{d}(n)-1)} \begin{cases} \psi_1 \sqrt{|T_n|} & \text{if } d < 2\alpha, \\ \psi_2 \log_2 |T_n| \sqrt{|T_n|} & \text{if } d = 2\alpha, \\ \psi_1 |T_n|^{1-\frac{\alpha}{d}} & \text{if } d > 2\alpha, \end{cases}
$$

Since this inequality holds for all pruning $\mathcal{T} \in \mathcal{P}(\mathcal{T}_0)$, one can take the infimum over all pruning in $\mathcal{P}(\mathcal{T}_0)$ to get the desired upper-bound:

$$
\operatorname{Reg}_T(f) \leqslant \inf_{\mathcal{T} \in \mathcal{P}(\mathcal{T}_0)} \left\{ \beta_1 \sqrt{|\mathcal{L}(\mathcal{T})|T} + \beta_2 |\mathcal{L}(\mathcal{T})| \right.
$$

$$
\left. + G|\mathcal{X}|^\alpha \sum_{n \in \mathcal{L}(\mathcal{T})} L_n(f) 2^{-\alpha(\mathrm{d}(n)-1)} \begin{cases} \psi_1 \sqrt{|T_n|} & \text{if } d < 2\alpha, \\ \psi_2 \log_2 |T_n| \sqrt{|T_n|} & \text{if } d = 2\alpha, \\ \psi_1 |T_n|^{1-\frac{\alpha}{d}} & \text{if } d > 2\alpha, \end{cases} \right\},
$$

with $\beta_1 = 2C_3 G \sqrt{\log\left(2BT|\mathcal{N}(\mathcal{T}_0)|\right)}$ and $\beta_2 = G(2^{-1}C_1 + C_2 2^{-1} T^{-\frac{1}{2}} + C_4)$.

• *Case 2:* $(\ell_t)_{1\leqslant t\leqslant T}$ $\eta$-*exp-concave.*

If the sequence of loss functions $(\ell_t)$ is $\eta$-exp-concave for some $\eta > 0$, then thanks to a Lemma in Hazan et al. (2016) we have for any $0 < \mu \leqslant \frac{1}{2}\min\{\frac{1}{G},\eta\}$ and all $t \geqslant 1, n \in \mathcal{L}(\mathcal{T}), k \in [K]$:

$$
\begin{aligned}
(\ell_t(\hat{f}_t(x_t)) - \ell_t(\hat{f}_{n,k,t}(x_t)))\mathbb{1}_{x_t\in\mathcal{X}_n} &\leqslant \big(\tilde{\mathbf{g}}_t^\top\tilde{\mathbf{w}}_t - \tilde{g}_{n,k,t} - \frac{\mu}{2}\big(\tilde{\mathbf{g}}_t^\top\tilde{\mathbf{w}}_t - \tilde{g}_{n,k,t}\big)^2\big)\mathbb{1}_{x_t\in\mathcal{X}_n} \\
&= \tilde{\mathbf{g}}_t^\top\mathbf{w}_t - \tilde{g}_{n,k,t} - \frac{\mu}{2}\big(\tilde{\mathbf{g}}_t^\top\mathbf{w}_t - \tilde{g}_{n,k,t}\big)^2 \qquad \leftarrow \text{ by (34)}.
\end{aligned}
$$
(38)

Summing (38) over $t \in [T]$ and $n \in \mathcal{L}(\mathcal{T})$, we get:

$$
\begin{aligned}
R_1 &\leqslant \sum_{n\in\mathcal{L}(\mathcal{T})}\sum_{t\in T_n}\tilde{\mathbf{g}}_t^\top\tilde{\mathbf{w}}_t - \tilde{g}_{n,k,t} - \frac{\mu}{2}\sum_{n\in\mathcal{N}(\mathcal{P})}\sum_{t\in T_n}\big(\tilde{\mathbf{g}}_t^\top\mathbf{w}_t - \tilde{g}_{n,k,t}\big)^2 \\
&\leqslant C_4 G|\mathcal{L}(\mathcal{T})| + \tilde{C}_3\sum_{n\in\mathcal{L}(\mathcal{T})}\sqrt{\sum_{t\in T_n}\big(\tilde{\mathbf{g}}_t^\top\mathbf{w}_t - \tilde{g}_{n,k,t}\big)^2} - \frac{\mu}{2}\sum_{n\in\mathcal{L}(\mathcal{T})}\sum_{t\in T_n}\big(\tilde{\mathbf{g}}_t^\top\mathbf{w}_t - \tilde{g}_{n,k,t}\big)^2 \quad \leftarrow \text{ by (36)},
\end{aligned}
$$
(39)

where we set $\tilde{C}_3 = C_3\sqrt{\log\big(K|\mathcal{N}(\mathcal{T}_0)|\big)}$. Young's inequality gives, for any $\nu > 0$, the following upper-bound:

$$
\sqrt{\sum_{t\in T_n}\big(\tilde{\mathbf{g}}_t^\top\mathbf{w}_t - \tilde{g}_{n,k,t}\big)^2} \leqslant \frac{1}{2\nu} + \frac{\nu}{2}\sum_{t\in T_n}\big(\tilde{\mathbf{g}}_t^\top\mathbf{w}_t - \tilde{g}_{n,k,t}\big)^2.
$$
(40)

Finally, plugging (40) with $\nu = \mu/\tilde{C}_3 > 0$ in (39), we get

$$
\begin{aligned}
R_1 &\leqslant C_4 G|\mathcal{L}(\mathcal{T})| + \tilde{C}_3\sum_{n\in\mathcal{L}(\mathcal{T})}\left(\frac{\tilde{C}_3}{2\mu} + \frac{\mu}{2\tilde{C}_3}\sum_{t\in T_n}\big(\tilde{\mathbf{g}}_t^\top\mathbf{w}_t - \tilde{g}_{n,k,t}\big)^2\right) - \frac{\mu}{2}\sum_{n\in\mathcal{L}(T)}\sum_{t\in T_n}\big(\tilde{\mathbf{g}}_t^\top\mathbf{w}_t - \tilde{g}_{n,k,t}\big)^2 \\
&= \left(\frac{C_3^2\log\big(K|\mathcal{N}(\mathcal{T}_0)|\big)}{2\mu} + C_4 G\right)|\mathcal{L}(\mathcal{T})|.
\end{aligned}
$$
(41)

To conclude, if $(\ell_t)$ are $\eta$-exp-concave, one has via (25), (30) and (41)

$$
\text{Reg}_T(f) \leqslant \beta_3|\mathcal{L}(\mathcal{T})| + G|\mathcal{X}|^\alpha\sum_{n\in\mathcal{L}(\mathcal{T})}L_n(f)2^{-\alpha(\mathrm{d}(n)-1)}\begin{cases}\psi_1\sqrt{|T_n|} & \text{if } d < 2\alpha, \\ \psi_2\log_2|T_n|\sqrt{|T_n|} & \text{if } d = 2\alpha, \\ \psi_1|T_n|^{1-\frac{\alpha}{d}} & \text{if } d > 2\alpha,\end{cases}
$$

with $\beta_3 = \frac{C_3^2\log\big(2BT|\mathcal{N}(\mathcal{T}_0)|\big)}{2\mu} + C_4 G + 2^{-1}G(C_1 + C_2 T^{-\frac{1}{2}})$, $0 < \mu < \frac{1}{2}\min\{\frac{1}{G},\eta\}$ and $\psi_1, \psi_2$ defined in (30). Again, taking infimum over $\mathcal{T} \in \mathcal{P}(\mathcal{T}_0)$ gives the result.

**Worst case regret bound**    Note that since we assume that $\|f\|_\infty \leqslant B$, and that all local predictors $\hat{f}_{n,k}, n \in \mathcal{N}(\mathcal{T}_0), k \in [K]$ in Algorithm 2 are clipped in $[-B, B]$, we first have for any $x \in \mathcal{X}$,

$$
|\hat{f}_t(x)| = \sum_{n\in\mathcal{N}(\mathcal{T}_0)}\sum_{k=1}^K w_{n,k,t}|\hat{f}_{n,k,t}(x)| \leqslant B\sum_{n\in\mathcal{N}(\mathcal{T}_0)}\sum_{k=1}^K w_{n,k,t} = B.
$$

Thus,

$$\begin{aligned}
\mathrm{Reg}_T(f) &= \sum_{t=1}^{T} \ell_t(\hat{f}_t(x_t)) - \ell_t(f^*(x_t)) \\
&\leqslant \sum_{t=1}^{T} G|\hat{f}_t(x_t) - f^*(x_t)| \qquad\qquad \leftarrow \ell_t \text{ is } G\text{-Lipschitz} \\
&\leqslant G \sum_{t=1}^{T} (|\hat{f}_t(x_t)| + |f^*(x_t)|) \\
&= 2BGT
\end{aligned}$$

(42)

$\blacksquare$

## Appendix C. Proof of Corollary 1

We state here a complete version of Corollary 1.

**Corollary 2** *Let $\alpha \in (0, 1], 1 \leqslant d \leqslant 2\alpha$. Under the same assumptions as in Theorem 2, Algorithm 2 achieves a regret with respect to any $f \in \mathscr{C}^\alpha(\mathcal{X}, L), L > 0$:*

$$\mathrm{Reg}_T(f) \lesssim \inf_{\mathcal{T} \in \mathcal{P}(\mathcal{T}_0)} \left\{ \sum_{n \in \mathcal{L}(\mathcal{T})} \min \left( 1 + L_n(f)|\mathcal{X}_n|^\alpha, \left( L_n(f)|\mathcal{X}_n|^\alpha \right)^{\frac{1}{2\alpha}} \right) \sqrt{T_n} \right\},$$

*where $\lesssim$ is a rough inequality that depends on $C_i$, $i = 1, \ldots, 4$ but is independent of $L, X, T$. Moreover, if $(\ell_t)$ are exp-concave:*

$$\mathrm{Reg}_T(f) \lesssim \inf_{\mathcal{T} \in \mathcal{P}(\mathcal{T}_0)} \left\{ \sum_{n \in \mathcal{L}(\mathcal{T})} \min \left( L_n(f)|\mathcal{X}_n|^\alpha \sqrt{|T_n|}, \left( L_n(f)|\mathcal{X}_n|^\alpha \right)^{\frac{2}{2\alpha+1}} |T_n|^{\frac{1}{2\alpha+1}} \right) \right\},$$

*where $\lesssim$ also depends on the exp-concavity constant.*

**Proof** [of Corollary 2]
We consider 2 cases:

1. *Case $d < 2\alpha$ (i.e. $d = 1, \alpha \in (\frac{1}{2}, 1])$.*

   Let $f \in \mathscr{C}^\alpha(\mathcal{X}, L)$ and $L > 0$ and fix any pruning $\mathcal{T} \in \mathcal{P}(\mathcal{T}_0)$. We will apply Theorem 2 to an extended pruning $\mathcal{T}'$, in which we extend each leaf $n \in \mathcal{L}(\mathcal{T})$ by a regular tree of depth $h_n \in \mathbb{N}$ to be optimized later in the proof. In particular, for each leaf $n$ in the original pruning $\mathcal{T}$, $\mathcal{T}'$ has $2^{h_n}$ leaves $m$ at depth $\mathrm{d}(m) = \mathrm{d}(n) + h_n \geqslant \mathrm{d}(n)$ with $L_m(f) \leqslant L_n(f)$. In particular, when $h_n = 0$, the original pruning $\mathcal{T}$ is recovered.

   (a) *Case $(\ell_t)$ convex:*

   Thanks to Theorem 2 (without applying Inequality (37) in the term depending on $C_3$), one has for $d = 1 < 2\alpha$:

   $$\begin{aligned}
   \mathrm{Reg}_T(f) &\leqslant 2C_3\sqrt{\log\left(K|\mathcal{N}(\mathcal{T}_0)|\right)} \sum_{m \in \mathcal{L}(\mathcal{T}')} \sqrt{|T_m|} + C_4 G|\mathcal{L}(\mathcal{T}_1)| \\
   &\quad + G\psi_1 \sum_{m \in \mathcal{L}(\mathcal{T}')} L_m(f)|\mathcal{X}_m|^\alpha \sqrt{|T_m|}, \\
   &\leqslant \min_{h_n \in \mathbb{N}} \left\{ C \sum_{n \in \mathcal{L}(\mathcal{T})} \left( \sqrt{2^{h_n}|T_n|} + 2^{h_n} + L_n(f)|\mathcal{X}_n|^\alpha 2^{-\alpha h_n} \sqrt{2^{h_n}|T_n|} \right) \right\}
   \end{aligned}$$
   (43)

   where $C > 0$ is some constant that depends on $C_3, C_4, G, X, \log(T), B$ and $\psi_1$ (defined in Theorem 2) but independent of other quantities $L_n(f), T, T_n$, that is used to simplify the presentation and may change from a display to another along the proof. Then, optimizing over $h_n$ so that

   $$\sqrt{2^{h_n}|T_n|} = L_n(f)|\mathcal{X}_n|^\alpha 2^{-\alpha h_n} \sqrt{2^{h_n}|T_n|},$$

we set

$$h_n = \max\left\{0, \frac{1}{\alpha}\log_2\left(L_n(f)|\mathcal{X}_n|^\alpha\right)\right\} \geqslant 0$$

which yields

$$\mathrm{Reg}_T(f) \leqslant C \sum_{n\in\mathcal{L}(\mathcal{T})} \min\left\{1 + L_n(f)|\mathcal{X}_n|^\alpha, \left(L_n(f)|\mathcal{X}_n|^\alpha\right)^{\frac{1}{2\alpha}}\right\}\sqrt{|T_n|}.$$

(b) *Case $(\ell_t)$ exp-concave:*

Since $(\ell_t)$ are exp-concave, Theorem 2 (with Inequality (41)) gives, for $d < 2\alpha$, for any extension $\mathcal{T}_1$ of pruning $\mathcal{T} \in \mathcal{P}(\mathcal{T}_0)$:

$$\mathrm{Reg}_T(f) \leqslant C\left(|\mathcal{L}(\mathcal{T}_1)| + \sum_{m\in\mathcal{L}(\mathcal{T}_1)} L_m(f)|\mathcal{X}_m|^\alpha\sqrt{|T_m|}\right), \tag{44}$$

$$\leqslant C \sum_{n\in\mathcal{L}(\mathcal{T})}\left(2^{h_n} + L_n(f)|\mathcal{X}_n|^\alpha 2^{-\alpha h_n}\sqrt{2^{h_n}|T_n|}\right), \tag{45}$$

where again $C > 0$ is a constant independent of $L$, $L_n(f)$, $|T_n|$ and $h_n$ that may change from a display to another. Optimizing over $h_n$ by equalizing the terms:

$$2^{h_n} = L_n(f)|\mathcal{X}_n|^\alpha 2^{-\alpha h_n}\sqrt{2^{h_n}|T_n|}$$

leads to

$$h_n = \max\left\{0, \frac{2}{2\alpha+1}\log_2\left(L_n(f)|\mathcal{X}_n|^\alpha|T_n|^{\frac{1}{2}}\right)\right\},$$

which yields and concludes the proof:

$$\mathrm{Reg}_T(f) \leqslant C \sum_{n\in\mathcal{L}(\mathcal{T})} \min\left\{L_n(f)|\mathcal{X}_n|^\alpha\sqrt{|T_n|}, \left(L_n(f)|\mathcal{X}_n|^\alpha\right)^{\frac{2}{2\alpha+1}}|T_n|^{\frac{1}{2\alpha+1}}\right\}$$

2. *Case $d = 2\alpha$.*

The proof is the same as for the case $d < 2\alpha$ but with $C$ now depending on $\psi_2$ (also defined in Theorem 1) rather than $\psi_1$. We get the same result.

■

## Appendix D. Proof of Equation (9)

One has, for any pruning $\mathcal{T} \in \mathcal{P}(\mathcal{T}_0)$ and some $f \in \mathscr{C}^\alpha(\mathcal{X}, L)$:

$$
\operatorname{Reg}_T(f) \lesssim \sum_{n \in \mathcal{L}(\mathcal{T})} \left(L_n(f)|\mathcal{X}_n|^\alpha\right)^{\frac{2}{2\alpha+1}} |T_n|^{\frac{1}{2\alpha+1}} = \sum_{n \in \mathcal{L}(\mathcal{T})} \left(L_n(f)^{\frac{1}{\alpha}}|\mathcal{X}_n|\right)^{\frac{2\alpha}{2\alpha+1}} |T_n|^{\frac{1}{2\alpha+1}}
$$

$$
\leqslant \left( \sum_{n \in \mathcal{L}(\mathcal{T})} L_n(f)^{\frac{1}{\alpha}}|\mathcal{X}_n| \right)^{\frac{2\alpha}{2\alpha+1}} \left| \sum_{n \in \mathcal{L}(\mathcal{T})} T_n \right|^{\frac{1}{2\alpha+1}}
$$

$$
= \left( \sum_{n \in \mathcal{L}(\mathcal{T})} L_n(f)^{\frac{1}{\alpha}}|\mathcal{X}_n| \right)^{\frac{2\alpha}{2\alpha+1}} |T|^{\frac{1}{2\alpha+1}}
$$

where inequality is obtained with Hölder's inequality with $p = (2\alpha + 1)/2\alpha$ and $q = 2\alpha + 1$. One could also write:

$$
\left( \sum_{n \in \mathcal{L}(\mathcal{T})} L_n(f)^{\frac{1}{\alpha}}|\mathcal{X}_n| \right)^{\frac{2\alpha}{2\alpha+1}} = \left( |\mathcal{X}| \sum_{n \in \mathcal{L}(\mathcal{T})} L_n(f)^{\frac{1}{\alpha}} \frac{|\mathcal{X}_n|}{|\mathcal{X}|} \right)^{\frac{2\alpha}{2\alpha+1}} := \left( |\mathcal{X}|^\alpha \|f\|_{\mathcal{L}(\mathcal{T}), \frac{1}{\alpha}} \right)^{\frac{2}{2\alpha+1}},
$$

where $f \mapsto \|f\|_{\mathcal{L}(\mathcal{T}), \frac{1}{\alpha}}$ is some $\frac{1}{\alpha}$-norm (or expectation) of $f$ over leaves $n \in \mathcal{L}(\mathcal{T})$ with probability $|\mathcal{X}_n|/|\mathcal{X}| = 2^{-\mathrm{d}(n)}, n \in \mathcal{L}(\mathcal{T})$.

## Appendix E. Comparison with Kuzborskij and Cesa-Bianchi (2020)

Let $f \in \mathscr{C}^\alpha(\mathcal{X}, L), (M^{(k)})_{1 \leqslant k \leqslant \mathrm{d}(\mathcal{T}_0)}$ such that $M^{(k)} \geqslant L_n(f)$ for any $n \in \mathcal{N}(\mathcal{T}_0), \mathrm{d}(n) = k$ and $T^{(k)} = \{1 \leqslant t \leqslant T : x_t \in \mathcal{X}_n, \mathrm{d}(n) = k\}$. Let $\mathcal{T}$ be any pruning and let $\alpha = 1, d = 1$. We have for the squared (exp-concave) loss, according to Corollary 1:

$$
\begin{aligned}
\mathrm{Reg}_T(f) &\lesssim \sum_{n \in \mathcal{L}(\mathcal{T})} \min \left\{ \left(L_n(f) \frac{|\mathcal{X}_n|}{|\mathcal{X}|}\right)^{\frac{2}{3}} |T_n|^{\frac{1}{3}}, \left(L_n(f) \frac{|\mathcal{X}_n|}{|\mathcal{X}|}\right)^{\frac{1}{2}} |T_n|^{\frac{1}{2}} \right\} \\
&= \sum_{k=1}^{\mathrm{d}(\mathcal{T})} \sum_{n \in \mathcal{L}(\mathcal{T}):\mathrm{d}(n)=k} \min \left\{ \left(L_n(f) \frac{|\mathcal{X}_n|}{|\mathcal{X}|}\right)^{\frac{2}{3}} |T_n|^{\frac{1}{3}}, \left(L_n(f) \frac{|\mathcal{X}_n|}{|\mathcal{X}|}\right)^{\frac{1}{2}} |T_n|^{\frac{1}{2}} \right\} \\
&\leqslant \sum_{k=1}^{\mathrm{d}(\mathcal{T})} \min \left\{ \left(\sum_{n \in \mathcal{L}(\mathcal{T}):\mathrm{d}(n)=k} L_n(f)2^{-k}\right)^{\frac{2}{3}} |T^{(k)}|^{\frac{1}{3}}, \left(\sum_{n \in \mathcal{L}(\mathcal{T}):\mathrm{d}(n)=k} L_n(f)2^{-k}\right)^{\frac{1}{2}} |T^{(k)}|^{\frac{1}{2}} \right\}
\end{aligned}
\tag{46}
$$

$$
\begin{aligned}
&\leqslant \sum_{k=1}^{\mathrm{d}(\mathcal{T})} \min \left\{ \left(M^{(k)} w^{(k)}\right)^{\frac{2}{3}} |T^{(k)}|^{\frac{1}{3}}, \left(M^{(k)} w^{(k)}\right)^{\frac{1}{2}} |T^{(k)}|^{\frac{1}{2}} \right\} \\
&\leqslant \min \left\{ \left(\bar{M}(f)\right)^{\frac{2}{3}} T^{\frac{1}{3}}, \left(\bar{M}(f)T\right)^{\frac{1}{2}} \right\}
\end{aligned}
\tag{47}
$$

where $\bar{M}(f) = \sum_{k=1}^{\mathrm{d}(\mathcal{T})} w^{(k)} M^{(k)}$ with $w^{(k)} = \sum_{n \in \mathcal{L}(\mathcal{T}):\mathrm{d}(n)=k} \frac{|\mathcal{X}_n|}{|\mathcal{X}|} = 2^{-k} |\{n \in \mathcal{L}(\mathcal{T}) : \mathrm{d}(n) = k\}|$ the proportion of leaves in $\mathcal{L}(\mathcal{T})$ at level $k$ in $\mathcal{N}(\mathcal{T}_0)$ and where we applied Hölder's inequality to get (46) and (47). The last upper-bound recovers and improves the one of Kuzborskij and Cesa-Bianchi (2020) for dimension $d = 1$, as described in the main part of the paper. For higher dimension $d \geqslant 2$, both for exp-concave and convex losses, Theorem 2 gives for any pruning $\mathcal{T}$ and any function $f \in \mathscr{C}^1(\mathcal{X}, L)$ (ignoring the dependence on $\mathcal{X}$):

$$
\begin{aligned}
\mathrm{Reg}_T(f) &\lesssim \sum_{n \in \mathcal{L}(\mathcal{T})} L_n(f)|T_n|^{1-\frac{1}{d}} = \sum_{k=1}^{\mathrm{d}(\mathcal{T})} \sum_{n \in \mathcal{L}(\mathcal{T}):\mathrm{d}(n)=k} L_n(f)|T_n|^{\frac{d-1}{d}} \\
&\leqslant \sum_{k=1}^{\mathrm{d}(\mathcal{T})} M^{(k)} |\{n : \mathrm{d}(n) = k\}|^{\frac{1}{d}} |T^{(k)}|^{\frac{d-1}{d}},
\end{aligned}
$$

which grows as $O\left(\sum_k M^{(k)} |T^{(k)}|^{\frac{d-1}{d}}\right)$ compared to $O\left(\sum_k (M^{(k)} |T^{(k)}|)^{\frac{d}{d+1}}\right)$ in Kuzborskij and Cesa-Bianchi (2020). As a consequence, if for every level $k = 1, \ldots, \mathrm{d}(\mathcal{T}), M^{(k)} |T^{(k)}|^{\frac{d-1}{d}} \geqslant (M^{(k)} |T^{(k)}|)^{\frac{d}{d+1}}$ it turns out that $M^{(k)} \geqslant |T^{(k)}|^{\frac{1}{d}}$ which leads to an equivalent bound $O(\sum_k |T^{(k)}|) = O(T)$ which corresponds to the worst case regret bound (42). As a conclusion, our bound recovers and improves their results in particular with a dependence to lower constants $L_n(f)$ and lower time rate $|T_n|^{1-\frac{1}{d}}$.

## Appendix F. Experiments

The following presents experimental results in a synthetic regression setting for both the Chaining Tree method (Algorithm 1) and the Locally Adaptive Online Regression (Algorithm 2. We consider the model $y_t = f(x_t) + \varepsilon_t$, where $\varepsilon_t \sim \mathcal{N}(0, \sigma^2)$ with $\sigma = 0.5$, $f(x) = \sin(10x) + \cos(5x) + 5$, for $x \in \mathcal{X} = [0, 1]$ and $\sup_x |f'(x)| \leqslant 15 =: L$. Furthermore, we assume that $x_t$ is independently drawn from the uniform distribution $\mathcal{U}(\mathcal{X})$.

Refer to Theorem 1, Theorem 2 and Corollary 1 in the paper to compare experimental results to theoretical guarantees. Figures can be reproduced using the code available on GitHub.

*Key observations:*

- For the squared loss, $\ell_t(\hat{y}) = (\hat{y} - y_t)^2$, Algorithm 2 achieves a time rate of $O(T^{\frac{1}{3}})$ compared to $O(\sqrt{T})$ for Algorithm 1 - see Figure 4(a). However, the trade-off is an increased dependence on the smoothness $L$, shifting from $O(L^{\frac{1}{2}})$ to $O(L^{\frac{2}{3}})$;
- We observe in Figure 5 that Algorithm 2 reduces regret with respect to $L$: it achieves $O(\sqrt{L})$ for absolute loss in Fig. 5(b) and $O(L^{\frac{2}{3}})$ for square loss in Fig. 5(a);
- In Figure 5(a) and Figure 5(b), we observe that both the experimental and theoretical curves level off once $L$ increases beyond a certain threshold $L_0 \gtrsim BT$. Indeed, we demonstrated that for any $f \in \mathscr{C}^{\alpha}(\mathcal{X}, L)$,

$$\text{Reg}_T(f) \lesssim \min\left\{ BT, \sum_n L_n(f)|T_n|^{1-\frac{1}{d}} \right\}.$$

See Appendix B and Equation (42) for more details.

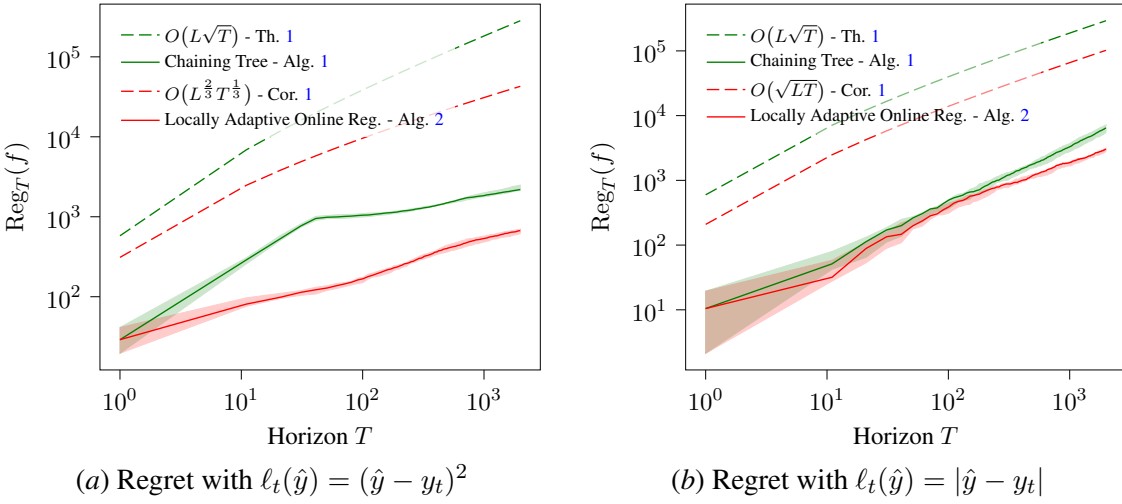

(a) Regret with $\ell_t(\hat{y}) = (\hat{y} - y_t)^2$        (b) Regret with $\ell_t(\hat{y}) = |\hat{y} - y_t|$

Figure 4: Comparison of regret as a function of $T$ for square and absolute loss functions. The dotted lines represent the theoretical results (where $O$ hides terms in $\log T$), while the solid lines show the actual performance of our algorithms (mean $\pm$ std over 5 runs).

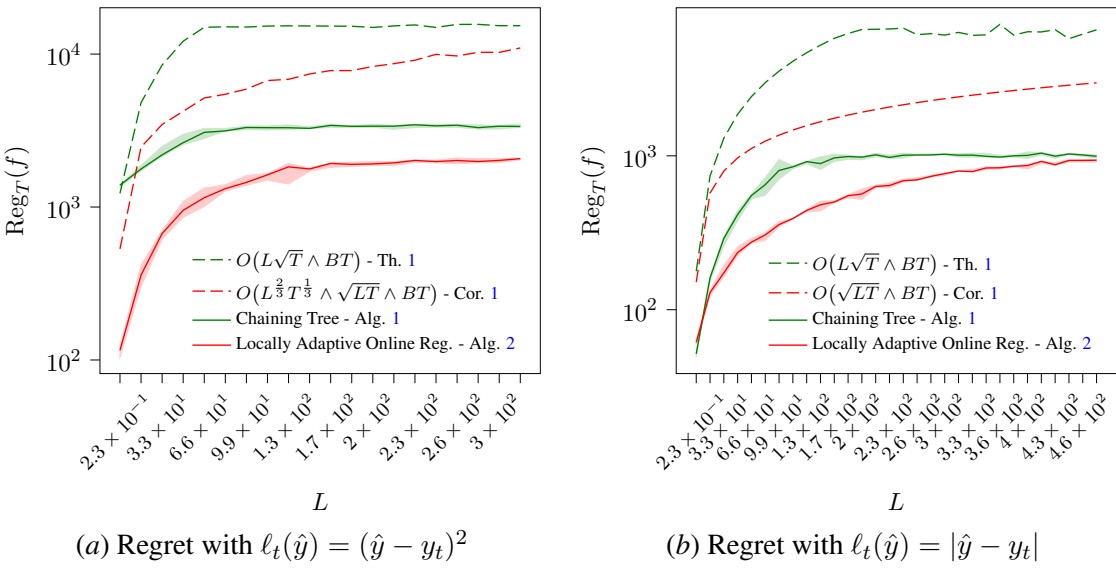

(a) Regret with $\ell_t(\hat{y}) = (\hat{y} - y_t)^2$   (b) Regret with $\ell_t(\hat{y}) = |\hat{y} - y_t|$

Figure 5: Regret (mean $\pm$ std over 5 runs) as a function of $L$ for square and absolute loss functions, with a fixed horizon of $T = 2000$. The analysis uses 20 equally spaced constants $l \in [2^{-6}, 2^5]$, which define the different Lipschitz functions where we apply our algorithms, given by $f_l(x) = f(lx)$ such that $\sup_{x \in \mathcal{X}} |f_l'(x)| \leqslant 15l =: L$.

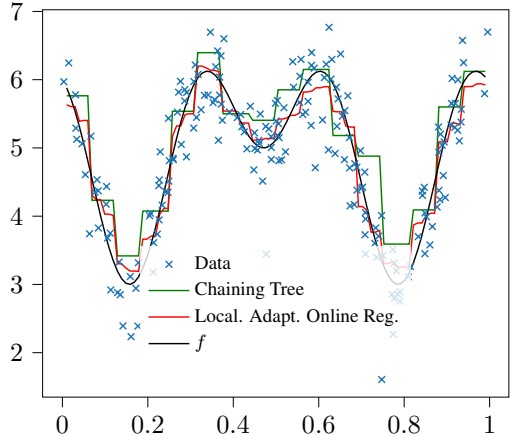

Figure 6: Predictions for Chaining Tree (Alg. 1) and Locally Adaptive Online Regression (Alg. 2) after $T = 1000$ data. For illustration purposes, we set the depth of the Chaining Trees to 5 and that of the Core Tree to 3.

*Note:* A minor adjustment has been made to the implementation of the Locally Adaptive Online Regression algorithm (Alg. 2). Rather than performing a grid search to determine the root nodes of the CT in Core Tree $\mathcal{T}_0$, we employ a *Follow the Leader* (or best expert) strategy. For squared losses, this method offers a similar benefit, that is reducing the regret for learning the root nodes from $O(B\sqrt{T})$ to $O(B \log(T))$ - see Cesa-Bianchi and Lugosi (2006, Chap 3.2). Consequently, the overall performance bound is improved, especially in low-dimensional cases (see Corollary 1, exp-concave case).

