# OpenReview forum: "Minimax-optimal and Locally-adaptive Online Nonparametric Regression"
_algorithmiclearningtheory.org/ALT/2025/Conference — ALT 2025_

### Official Review · Reviewer_Nj92 · 2024-11-08
**Solid paper on gradient boosting for nonparametric regression.**

**Rating:** 7
**Confidence:** 2

**Review:**

Summary of the paper:
This paper introduces an online gradient-boosting algorithm (Algorithm 1) and regret guarantees (Theorem 1) that do not require prior knowledge of $L$ and $\alpha$. This result is achieved via a chaining tree and is minimax optimal. Section 3.3. extends these results in an adaptive-boosting style. Algorithm 2 is able to adapt to local curvature using a 'sleeping expert' approach.
Detailed proofs and experiments are provided in the appendix.


Questions:
(Q1): Theorem 1 guarantees a regret bound of $T^{1-\frac{\alpha}{d}}$ and at the beginning of Section 3.3, it is claimed that the regret is in the order of $T^{\frac{d-1}{d}}$. I can not see how a regret bound of $T^{\frac{d-1}{d}}$ follows from Theorem 1 if $\alpha$ is small compared to $d$. E.g., if $d = 4\alpha$. What am I missing?

Strengths:

(S1): mostly clearly written.

(S2): the paper provides a nice insight that parameter-free weak learners imply parameter-free strong learners.

(S3):  it provides a nice result by combining several existing results (regret bounds on parameter-free online learning) and techniques (chaining trees, sleeping experts).

(S4): The proofs are clearly written and, to the best of my understanding, correct.

Weaknesses:

(W1): It would be helpful to state Theorem 2 and Corollary 1 in a more formal way and state the constants explicitly. Specifically, the 'rough inequalities' are confusing to me.

(W2):  To the best of my understanding, the results are primarily achieved by an interesting combination of existing results and techniques and not by new technical tools and approaches.



I am a bit undecided about this paper. On the one hand, I can see its merits (S1- S4), on the other hand, to the best of my understanding the main contribution is that 'parameter-free' can be preserved through boosting. This result seems to the best of my understanding not too surprising. I am happy to revise my review if I am missing something.


%%%%%%%%%%%%% after rebuttal %%%%%%%%%%%

I thank the authors for their feedback and clarifications. Apologies for the slow reply from my side!

Answer to Answer on (W1): I think the authors should find a way to state Thm. 2 in a more formal way and avoid the 'rough inequalities'. While I can see the motivation for streamlining the discussion, these informal statements are confusing to me. (only a suggestion: maybe state the Thm. wrt a constant C>0 and define C in the appendix?)

Answer to Answer on (W2): I see your point.

Due to the answers and clarifications, I decided to raise my score.

**Paper Award:**

No

---

> ### Author Response · Authors · 2024-11-20
>
> Dear Reviewer, thank you for your feedback.  We appreciate the time and effort you put into reviewing our work, and we address your points in detail below.
>
> **Q1**: Thank you for pointing out this typo, $O(T^{(d-1)/d})$ is the specific minimax regret for Lipschitz functions ($\alpha = 1, d > 2$). The correct optimal regret is in $O(T^{(d-\alpha)/d})$ when competing against Hölder functions with smoothness $\alpha > 0, d > 2\alpha$, as established in Rakhlin and Sridharan [2015].
>
> **W1**: We chose to postpone the complete version of this Theorem and Corollary (including exact constants) in Appendix B and Appendix C respectively. Indeed, we were concerned that including a detailed presentation in the main body of the paper might hinder readability and understanding. Therefore, we chose to present the results in a way that highlights only the dependencies affecting the final regret rate in $T$ and $L$. Nevertheless, if the reviewers believe it would be more appropriate, we are open to incorporating the complete results into the main text.
>
> **W2**: We believe our contributions are technically significant and go beyond a straightforward combination of existing methods. Specifically:
> - While the chaining tree structure has been utilized in previous work, such as Rakhlin and Sridharan [2014], [2015], our approach is distinct in that it provides a constructive analysis. In comparison to Gaillard and Gerchinovitz [2015], our guarantees apply to general convex losses and incorporate local dependencies on the competitor function, without requiring it beforehand.
> - Our $\texttt{LocAdaBoost}$ algorithm achieves a dual adaptation through the use of a pruning process, allowing it to adapt to both the local profile of the target Hölder function and the curvature of the loss. To the best of our knowledge, we are the first to propose a computationally feasible algorithm that demonstrates such remarkable properties. As stated in Corollary 1, our algorithm achieves the best of both worlds, making the optimal compromise between:
>    - Adapting to regions of low variation in the target function with a dependence tailored to local regularities,
>    - Leveraging the curvature of the losses to accelerate the learning process in $T$.
> - Additionally, we highlight that merely employing 'parameter-free' on a chaining tree (as discussed below Theorem 1) is insufficient to achieve optimal regret when dealing with losses that exhibit properties beyond basic convexity.
>
> Moreover, we believe our work offers valuable contributions to the community beyond its technical aspects:
> - We present a novel meta-algorithm for boosting adapted to the adversarial online setting. Even though $\texttt{OGB}$ is explicitly defined and rigorously analyzed using a chaining tree structure, we introduce it in a very general form, as we strongly believe it has the potential for further exploration in future work with various types of weak learners, such as shallow networks, splines, or other tree-based methods commonly used in the boosting and regression literature.
> - We formalized the learning trade-off between 1) adapting to the local profile of the target function and 2) achieving fast rates in $T$. This addresses a key open question raised in Kuzborskij and Cesa-Bianchi [2020] (see the end of Introduction), which inquired whether fast asymptotic rates of $T^{(d−1)/d}, \alpha=1, d>2$, improving upon their rates ($T^{d/(d+1)}$), could be achieved while maintaining local adaptivity. Our work provides an affirmative answer to this question and paves the way for developing a minimax theory that considers rates in both $L$ and $T$, rather than focusing solely on fast rates in $T$.
>
> If needed, we are happy to offer further clarifications or adjustments to better emphasize these aspects of our work. We appreciate your comments and the opportunity to address them.

---

### Official Review · Reviewer_BB1n · 2024-11-08

**Rating:** 7
**Confidence:** 4

**Review:**

This paper introduces a novel parameter-free online gradient boosting (OGB) algorithm for adversarial nonparametric regression with convex losses. By leveraging chaining trees and adaptive pruning, the algorithm achieves minimax optimal regret, adapting to local Lipschitz patterns. The approach is theoretically sound and innovative, expanding classical boosting to an adversarial online setting with computational efficiency. However, the paper’s dense theoretical presentation may limit accessibility, and the experimental validation is confined to an appendix. To improve the paper, consider simplifying the mathematical explanations, consolidating related work, and expanding on empirical results and real-world applications in the main text. Adding visual aids and a brief practical discussion would further enhance readability and impact.

The OGB method proposed in this paper is targeting optimal minimax regret performance with a focus on local adaptivity. The contribution is excellent and technically novel. Also, leveraging the concept of chaining trees and pruning based on local Lipschitz profiles is both innovative and well-grounded in nonparametric regression theory. To sum up, it is a theoretically sound, novel theoretical work.

**Paper Award:**

No

---

> ### Author Response · Authors · 2024-11-20
>
> Dear Reviewer,
>
> Thank you for your encouraging and positive feedback. We greatly appreciate your thoughtful suggestions for clarifications and simplifications and will incorporate them to enhance the overall quality and clarity of the paper in its final version.
> Regarding your suggestion to include additional experiments on real-world datasets, we acknowledge its potential to further validate our approach. While our current focus remains on synthetic datasets to illustrate the local adaptivity (with respect to $L$) of our $\text{LocAdaBoost}$ method, exploring real-world datasets is a promising direction that we could consider.

---

### Official Review · Reviewer_hZXz · 2024-11-12
**An interesting approach to online nonparametric regression based on an adaptive boosting algorithm**

**Rating:** 7
**Confidence:** 3

**Review:**

This paper concerns online nonparametric regression with general convex losses, where the comparator class of functions is Lipschitz. The authors employs an adaptive boosting algorithm in a specialized class of trees, termed "chaining trees", where the parameter updates can be computed efficiently due to a sparse structure of the gradients. By applying recent advances in adaptive online learning, the paper achieves several significant results:

1) The authors present a new and general online gradient boosting method, which can be performed efficiently within the class of chaining trees (in fact, the algorithm is defined more generally on some class of functions \mathcal{W})
2) The algorithm achieves minimax regret over the Hölder class of functions and adapts to unknown regularity constants (L or \alpha).
3) The authors also propose a meta-algorithm that competes with oracle-pruned trees, yielding optimal locally adaptive regret that scales with the local regularity of the function class and adapts to loss curvature (e.g., exp-concavity).

Overall, I liked the paper and I believe its theoretical contributions are substantial. The presentation is generally clear and the technical parts seem sound, up to what I was able to verify (only part of the proofs in the appendix). However, the paper is quite dense in math at some parts and it would benefit from more intuitive explanations. For instance, while the introduction of the chaining trees (Definition 1) was very clear, I was missing some discussion in the construction of locally adaptive boosting in Section 3.3.

Nevertheless, this paper is the first to present an efficient algorithm achieving locally adaptive regret relative to any Hölder class, without requiring knowledge of the local regularities or the loss curvature (while adapting to both). I would like to see the paper presented at the conference and so I recommend it for acceptance.

Some more remarks:

1) The online gradient boosting resembles an adaptive version of online gradient descent in online convex optimization (with parameter vector \theta). The analysis includes an additional step to bound the approximation error between the target Hölder function and the closest (in terms of cumulative loss) parameter \theta \in R^N, and uses the tree structure to bound the norm of \theta. Could you elaborate on the main novelties and challenges in your analysis compared to existing adaptive OCO results?

2) To be honest, I found the description of the locally adaptive boosting method somewhat challenging:
- When you say decision trees are "sitting in nodes of a core tree", does this mean they share a (sub)tree structure with the core tree?
- What role does each tree indexed by k=1,...,K play in the analysis. It seems that the only difference is that each tree at node n is initialized with a different \gamma_k.
- Could you clarify the node experts h_{l,t} as described in Algorithm 2? Are they are shared across different values of n and k (they are not indexed with n and k explicitly)?

3) How feasible is the method in high dimensions? It seems the computational complexity scales exponentially with dimension.

**Paper Award:**

No

---

> ### Author Response · Authors · 2024-11-20
>
> Dear Reviewer, we first thank you for your nice and constructive feedback. In particular, your observations have helped us identify areas where the exposition could be improved.
>
> **1) About the Locally Adaptive section 3.3**
>
> As you pointed out, our Locally Adaptive Boosting method is quite complex. Although we included explanatory graphs, we found it challenging to present it with the clarity we had aimed for. We will consider ways to make it more accessible in the final version (e.g. improving the graph, adding details) and are open to further feedback to enhance the presentation. Below are our responses to your specific points about this part:
>
> - *About "sitting in the nodes of the core tree".* We mean the following. First, the core tree partitions the input space $\mathcal X$ into  subsets $(\mathcal X_n)$. Then, at each node $n$ of the core tree, $K$ instances of the chaining tree are launched, indexed by $(n,k)$. The chaining trees sitting in a same node $n$ of the core tree share the same subregion $\mathcal X_n$ but is initialized at different values $\gamma_k \in \Gamma$.  Let $N$ be the number of nodes in the core tree, the final prediction of the procedure relies on combining the $N \times K$ predictions produced by the chaining trees, using an expert advice algorithm;
>
> - *About the definition of index $(n,k)$.*
> You are right: each chaining tree $(n,k)$ is launched over the sub region $\mathcal X_n$ whose root node is set to $\gamma_k$, on the grid $\Gamma$. It is key in our analysis because it allows the parameter-free subroutine of the chaining tree to be initialized on a value close to the optimal one (at most at a distance $(\gamma\_{k+1} - \gamma\_{k})/2$, the precision of the grid $\Gamma$). Indeed, learning the optimal initial value $\gamma_k$ turns out to be more efficient when using the expert advice algorithm rather than directly with the parameter-free subroutine;
>
> - *Clarification about $h\_{l,t}$, line 10, Alg. 2.*
> First, note that indeed it implicitly depends on both $n$ and $k$ through $l$. Here, $l$ refers to a node of the chaining tree $(n,k)$ and thus depends itself on $(n,k)$. We omitted the explicit dependence to simplify the notation, but we can reintroduce it if you feel it would enhance clarity. Specifically, there are no shared parameters between the chaining trees $(n,k)$. They work as independent subroutines running in parallel;
>
> **2) About complexity**
>
> This is a key strength of our algorithms: perhaps surprisingly, our space and time complexities do not increase with dimension. Intuitively, as the optimal regret deteriorates with higher $d$, the algorithm requires less precision, allowing it to utilize shallower trees with reduced depth of $O(\log(T)/d)$. This is detailed in the 'Complexity' paragraphs following Theorem 1 and Figure 3. The overall complexities after $T$ iterations of our algorithms can be summarized as follows:
> | Algorithm         | Time Complexity  | Space Complexity    |
> |-------------------|---------------------|---------------------|
> | Alg. 1  - *Chaining Tree*    | $O(T\times \frac{1}{d}\log(T))$      |  $O(T)$  |
> | Alg. 2  - *LocAdaBoost*  |  $O(T \times \frac{\sqrt{T}}{d^2}\log(T)^2)$    | $O(T^{3/2})$ |
>
>
> **3) About the resemblance to an adaptive OGD**
>
> If $N$ is the number of nodes in the chaining tree, applying an adaptive version of OGD to a 'global' parameter $\theta = (\theta_n) \in \mathbb{R}^N$ would result in a regret bound of $O(||\theta||\sqrt{\sum_t ||g_t||^2}) = O(\sqrt{T} \sum_n |\theta_n|)$, where the gradients satisfy $0 < ||g_t||$ at each time step. In contrast, our procedure considers a node-specific descent, yielding for each node $n$ a regret upper bound of $O(|\theta_n|\sqrt{\sum_t \|g_{n,t}\|^2})$, where $g_{n,t}$ is the gradient with respect to node $n$. Notably, $g_{n,t} = 0$ when the data $x_t$ does not fall in the corresponding cell of node $n$. This leads to an overall smaller regret scaling as $O(\sum_n |\theta_n|\sqrt{|T_n|}) $, where $T_n$ is the set of time steps for which $g_{n,t} \neq 0$.
>
> The challenge lies also in controlling the norms of the $\theta_n$ values by defining an appropriate structure and a suitable oracle that achieves low approximation error while maintaining small norms (ensuring small individual regret) as we are going through more refined subregion of the target Hölder function.

---

### Meta-Review · Area_Chair_bTvc · 2024-12-09

**Recommendation:** Accept
**Confidence:** 4

**Metareview:**

This paper introduces a novel parameter-free online gradient boosting (OGB) algorithm for adversarial nonparametric regression with general convex losses. By leveraging the innovative concept of "chaining trees" and adaptive pruning, the proposed approach achieves minimax optimal regret while adapting to local Lipschitz patterns. The theoretical contributions were universally recognized as both substantial and innovative by the reviewers, with particular praise for the efficient adaptation to unknown regularity constants and loss curvature.

Key strengths highlighted by the reviewers include:

1) The introduction of a computationally efficient algorithm that advances classical boosting into an adversarial online learning framework.
2) A minimax regret over the Hölder class and adaptability to local Lipschitz regularity without requiring prior knowledge.
3) The novel incorporation of chaining trees and pruning mechanisms, which are well-grounded in nonparametric regression theory.

The reviewers were unanimous in recognizing the paper's strong theoretical foundation and significant contributions to online learning and boosting algorithms. Based on the reviews and my own reading, I recommend the paper for acceptance.

**Paper Award:**

No